

# Human amplified changes in precipitation-runoff patterns in large river basins of the Midwestern United States

Sara A. Kelly[1], Zeinab Takbiri[2], Patrick Belmont[1], Efi Foufoula-Georgiou[3]

[1]Department of Watershed Sciences, Utah State University, Logan, 84322-5210, USA
[2]Department of Civil, Environmental, and Geo-Engineering, University of Minnesota, Minneapolis, 55455-0116, USA
[3]Department of Civil and Environmental Engineering, University of California, Irvine, CA 92697, USA

*Correspondence to*: Sara A. Kelly (sara.kelly@aggiemail.usu.edu)

**Abstract.** Complete transformations of land cover from prairie, wetlands, and hardwood forests to row crop agriculture and urban centers are thought to have caused profound changes in hydrology in the Upper Midwestern US since the 1800s. In
this study, we investigate four large (23,000-69,000 km$^2$) Midwest river basins that span climate and land use gradients to understand how climate and agricultural drainage have influenced basin hydrology over the last 79 years. We use daily, monthly, and annual flow metrics to document streamflow changes and discuss those changes in the context of precipitation and land use changes. Since 1935, flow, precipitation, artificial drainage extent, and corn and soybean acreage have increased across the region. In extensively drained basins, we observe 2 to 4 fold increases in low flows and 1.5 to 3 fold
increases in high and extreme flows. Using a water budget, we determined that the storage term has decreased in intensively drained and cultivated basins by 30%-200% since 1975, but increased by roughly 30% in the less agricultural basin. Storage has generally decreased during spring and summer months and increased during fall and winter months in all watersheds. Thus, the loss of storage and enhanced hydrologic connectivity and efficiency imparted by artificial agricultural drainage appear to have amplified the streamflow response to precipitation increases in the Midwest. Future increases in precipitation
are likely to further intensify drainage practices and increase streamflows. Increased streamflow has implications for flood risk, channel adjustment, and sediment and nutrient transport and presents unique challenges for agriculture and water resource management in the Midwest. Better documentation of existing and future drain tile and ditch installation is needed to further understand the role of climate versus drainage across multiple spatial and temporal scales.

# 1 Introduction

## 1.1 Whether humans, climate or both have caused streamflow change matters for water quality and watershed management

The magnitude and frequency of streamflows strongly influence water quality, sediment and nutrient transport, channel morphology, and habitat conditions of a river channel. While streamflows fluctuate naturally over event to





millennial timescales, humans have also altered rainfall-runoff processes in pervasive and profound ways (Vörösmarty et al., 2004). For example, humans have substantially altered the timing and magnitude of evapotranspiration, have dammed, channelized and leveed waterways, and have installed artificial drainage networks in former wetlands (Boucher et al., 2004; Dumanski et al., 2015; Rockström et al., 2014; Schottler et al., 2014; Vörösmarty et al., 2004) . While it is inevitable that

wetland removal and artificial drainage will change rainfall-runoff processes, the effects of drainage on the hydrologic cycle may be subtle and difficult to discern, and may manifest differently at different spatial scales and times of year (e.g., Bullock and Acreman, 2003; Foufoula-Georgiou et al., 2016; Irwin and Whiteley, 1983; O'Connell et al., 2007).

Systematic increases in peak, mean, total, and base flows are widely reported in the Midwestern USA and attributed to changes in climate, such as increasing precipitation and earlier snowmelt, and land use, including widespread conversion

from perennial vegetation, such as grasses, to annual row crops, primarily corn and soybean, and the addition of artificial drainage (e.g. Foufoula-Georgiou et al., 2015; Frans et al., 2013; Gerbert and Krug, 1996; Juckem et al., 2008; Novotny and Stefan, 2007; Schilling and Libra, 2003; Schottler et al., 2014; Zhang and Schilling, 2006). However, the question remains: how have climate and land use changes affected streamflows in very large ($>10^4$ km$^2$) watersheds, the scale at which many states and federal programs are often tasked with monitoring and evaluating water quality?

Many basins across the Midwestern corn belt and around the world are experiencing greater runoff, higher sediment and nutrient loads, and accelerated loss of habitat than in the past (Blann et al., 2009). For waters impaired by sediment under the US Clean Water Act (CWA), EU Water Framework Directive, and similar regulations around the world, loads often consist of both natural and human-derived sediment sources (Belmont et al., 2011; Gran et al., 2011). Differentiating between these two sources is often very difficult, and yet essential for identifying and achieving water quality standards

(Belmont et al., 2014; Trimble and Crosson, 2000; Wilcock, 2009). Sediment sources derived from near or within the channel itself (e.g., bank erosion from channel widening) are particularly sensitive to changes in streamflows (Lauer et al., in review; Schottler et al., 2014; Lenhart et al., 2013). Bank erosion is a significant sediment source in many alluvial rivers, contributing as much as 80% to 96% of the sediment that comprise a river's total sediment load (Kronvang et al., 2013; Palmer et al., 2014; Schaffrath et al., 2015; Simon et al., 1996; Stout et al., 2014; Willett et al., 2012). For some agricultural

basins, erosion of near-channel sources contributes more fine sediment than does agricultural field erosion (Belmont et al., 2011; Lenhart et al., 2012; Trimble, 1999). However, if artificial drainage practices act to amplify streamflows, then the source of accelerated bank erosion may still be linked to agriculture. Artificial drainage is currently unregulated at the federal level in the US and many countries around the world. Therefore, in stark contrast to urban hydrology, progress in understanding the effects of agricultural drainage has been hindered by the fact that accurate data regarding the location,

size, depth, efficiency and connectivity of sub-surface drainage systems are rarely available.





## 1.2 Artificial drainage improves agricultural productivity but may amplify streamflows in large watersheds

The United States is the largest producer of corn and soybeans in the world (Boyd and McNevin, 2015; Guanter et al., 2014). Exceptionally high agricultural productivity over the past century and a half required massive conversion of grasslands, wetlands, and forests to agricultural lands (Dahl, 1990; Dahl and Allord, 1996; Marschner, 1974). Although

many advances in cropping practices have led to the modern day prosperity of the Corn Belt, artificial drainage has played a critical role for agriculture in the Midwestern USA. Throughout this paper "artificial drainage" is used as a general term that refers to both human installed surface ditches and subsurface tile drainage. Tile drains and ditch networks are installed to ameliorate water-logged soils, which are known to limit crop growth (Hillel, 1998; Sullivan et al., 2001; Wuebker et al., 2001). Modern tile drains are composed of corrugated plastic tubing and are typically installed at depths of 1-2 m to control

the elevation of the water table below the soil surface (Hillel, 1998).

The economic benefits of artificial drainage are well understood by Midwestern farmers, who have invested heavily in drainage systems to reduce soil moisture, surface overland flow, and soil erosion, and increase land value, ease of equipment operation, and production of first class crops such as corn and soy (Burns, 1954; Fausey et al., 1987; Hewes and Frandson, 1952; Johnston, 2013; McCorvie and Lant, 1993). Installation or enhancement of tile drainage systems often

occurs simultaneously with land conversion from wild hay and small grains to soybeans as Fig. S1 demonstrates in the Supplement (Blann et al., 2009; Burns, 1954; Hewes and Frandson, 1952). Although the conversion of perennial grasses to corn and soybean rotations doesn't necessarily lead to a reduction in evapotranspiration (ET) over the course of an entire growing season, at least for well drained soils (Hamilton et al., 2015), several studies report a reduction of ET early in the growing season (Hickman et al., 2010; McIsaac et al., 2010; Schottler et al., 2014; Zeri et al., 2013). Thus changes in land

cover (and ET) and drainage expansion have been found to alter watershed hydrology and increase mean annual flows (Harrigan et al., 2014; Kibria et al., 2016), base flows (Juckem et al., 2008; Robinson, 1990; Schilling and Libra, 2003), annual peak flows (Dumanski et al., 2015; Magner et al., 2004; Skaggs et al., 1980, 1994), and total flow volumes (Dumanski et al., 2015; Frans et al., 2013; Lenhart et al., 2011). While it seems inevitable that altering ET and subsurface drainage efficiency should have measureable effects on streamflow, the combined effects have proven difficult to isolate

empirically, especially across scales, due to measurement uncertainties, high temporal and spatial variability in antecedent moisture conditions and runoff processes, a shift towards a wetter climate today than in the historical past, as well as limited documentation of artificial drainage installation in the US.

In this paper we couple analysis of historical patterns in large ($>10^4$ km$^2$) river basin hydrology in the Midwestern USA with historical climate and land use data to identify how each of these factors have influenced streamflow patterns.

Specifically, we address the following questions: (1) how have land use, climate, and streamflows changed during the 20$^{th}$ and 21$^{st}$ centuries; (2) what are the timing, time scales and times of year that changes are most prominent; and (3) can changes in climate alone explain changes in streamflow? We hypothesize that in the most intensively managed agricultural



basins, climate alone cannot explain streamflow patterns, and that land use changes in the Midwestern USA have amplified the expected hydrologic change associated with climate. We test this hypothesis in four large river basins with different histories and climates using a suite of quantitative methods that test the statistical significance of changes in streamflow and precipitation at multiple time scales. Finally, we present a water budget for each basin.

We acknowledge that the conversion of precipitation to streamflow occurs by a complex suite of physical processes. Inevitably, we lack temporal and spatial coverage/resolution of all of the relevant hydrologic fluxes (e.g., groundwater, actual evapotranspiration, infiltration, soil water flux rates) to characterize the system completely and have limited ability to ascribe subtle changes to any given physical process, especially at large scales. Yet, with increasing concerns about water quality and aquatic biota, disentangling the effects of artificial drainage and changing precipitation patterns is important for

evaluating economic costs, benefits and risks, predicting the effects of future land and water management and informing future policy.

## 2 Study areas: large river basins of the Midwest with varying degrees of climate and land use change

We analyze hydrologic and land use change in four large Midwestern watersheds during 1935-2013. We selected these basins for the following reasons: all are agricultural, to various degrees, primarily producing corn and soybeans; all are

located mainly within the Central Lowland physiographic province and were affected by continental glaciation resulting in mostly flat, poorly drained uplands and incised river valleys (Arnold et al., 1999; Barnes, 1997; Belmont et al., 2011; Day et al., 2013; Gran et al., 2009; Groschen et al., 2000; Rosenberg et al., 2005; Stark et al., 1996); and all are characterized by a humid, temperate climate (Kottek et al., 2006). Additionally, all four basins also contain waters impaired for excessive sediment under the US Clean Water Act. Therefore, deconvolving climate and land use effects on basin hydrology is

essential for developing and attaining sediment- and nutrient-related water quality standards. Despite the broad similarities between basins, we have intentionally selected watersheds that span a gradient of climate and land use change. From northwest to southeast, these include: the Red River of the North basin (RRB), upstream of Grand Forks, ND (67,005 km$^2$), Minnesota River basin (MRB), upstream of Jordan, MN (42,162 km$^2$), Chippewa River basin (CRB), upstream of Durand, WI (23,444 km$^2$), and Illinois River basin (IRB), upstream of Valley City, IL (69,268 km$^2$) (Fig. 1).

There is a broad northwest to southeast precipitation and temperature gradient across the region (Fig. S2). The RRB is the coldest and driest of all four study basins, although the last two decades (1990's and 2000's) have been the wettest in historical times. Precipitation records, lake level elevations, and paleoclimate studies indicate that the basin is prone to extreme climate variability (Fritz et al., 2000; Miller and Frink, 1984). Much like the RRB, the adjacent MRB is uniquely situated at a "climatic triple junction" where warm moist air from the Gulf of Mexico, cold dry air from the Artic, and dry

Pacific air dominate at different times of the year and have varied in relative dominance in the past (Dean and Schwalb, 2000; Fritz et al., 2000). Temperature and humidity in the CRB are more strongly influenced by the Great Lakes than in the



other basins. The southwest IRB generally receives more precipitation than the northeast in all months. On average each basin from northwest to southeast receives 589 mm, 716 mm, 822 mm, and 960 mm annually, with 59%-68% of the annual precipitation falling in the spring (MAM) and summer (JJA) months based on PRISM annual long term means, 1981-2010 (Fig. S2). Recent increases in precipitation and streamflows have been reported across the region during the last few decades

(Foufoula-Georgiou et al., 2015; Frans et al., 2013; Gerbert and Krug, 1996; Groisman et al., 2001; Juckem et al., 2008; Novotny and Stefan, 2007; Schottler et al., 2014).

Settlement, agricultural intensification, and development differ in timing and intensity among basins but are generally similar. During the early to mid-nineteenth century, permanent occupation of the Midwest was difficult without the aid of artificial drainage (Beauchamp, 1987). Beginning in the mid-1800s, organized drainage districts and enterprises

installed ditches and tile to drain many permanently or seasonally wet areas and create more arable land (Beauchamp, 1987; Skaggs et al., 1994). Between 1850 and 1930 Illinois, Minnesota, and Wisconsin lost an estimated 90%, 53%, and 32% of state wetlands, respectively (McCorvie and Lant, 1993). Enormous tracts of wetlands and tall grass prairie (millions of acres) were levelled and drained, mainly by surface ditches and canals, in the RRB during this same time (Miller and Frink, 1984). Artificial drainage increased property value, and as corn and soybean commodity prices increased, as they did following

WWII, in the mid-1970's, and most recently a tripling of commodity prices between 2002-2012 (Glaser, 2016; Johnston, 2013), lands previously cultivated for small grains or left as wet meadows were drained and converted to soybean and corn fields (Blann et al., 2009; Burns, 1954; Wright and Wimberly, 2013). Although many advances in cropping practices have led to the modern day prosperity of the Corn Belt, drainage installation and intensification has played a critical role for agriculture in the Midwestern US. Today the RRB, MRB, CRB, and IRB respectively contain 45%, 78%, 12% and 60% of

land cultivated for corn and soybeans, yet estimates of tile drainage in these basins remain poorly constrained (Fig. 1). Within the Bois de Sioux watershed, a sub-basin of the RRB where permits are required for drain tile installation, installation has increased from 3 miles in 1999 to 1,924 miles in 2015 for a cumulative total of 15,102 miles of new tile installed since 1999 (Bois de Sioux Watershed District, 2015). Tile drainage installation in all basins continues to this day.

The other major anthropogenic impact that affects all basins is dams installed for hydropower, navigation, water

resources, and recreation. Most of the dams in our study basins are small and were constructed in the late 1800's and early 1900's (Barnes, 1997; Delong, 2005; Graf, 1999; Hyden, 2010; Lian et al., 2012; Martin, 1965; Stoner et al., 1993; United States Army Corps of Engineers, 2016). Therefore, the effects of these dams would have been established well before our study period. For example, in the Illinois River basin all major dams had been completed by 1939. Based on work by Lian et al. (2012), streamflow changes post 1938, specifically peak flows, have been influenced more by climate than dam

operations, though they did not consider the effects of drain tile. One exception might be the uppermost Illinois River basin, which has been influenced by expansion of the Chicago metropolitan area. Though historical and present water withdrawals are largely unknown, increased water use for industry, agriculture, and public drinking supply may offset some of the




climate impacts of increased precipitation. Urban and suburban detention basins may also limit how much precipitation is converted to runoff. We expect that other water development projects in each basin have minimally affected streamflows at the basin outlet. Conversion of hay and small grains to corn and soybeans accompanied by artificial drainage expansion are likely the largest land use land cover (LULC) changes in these basins since the early to mid-twentieth century.

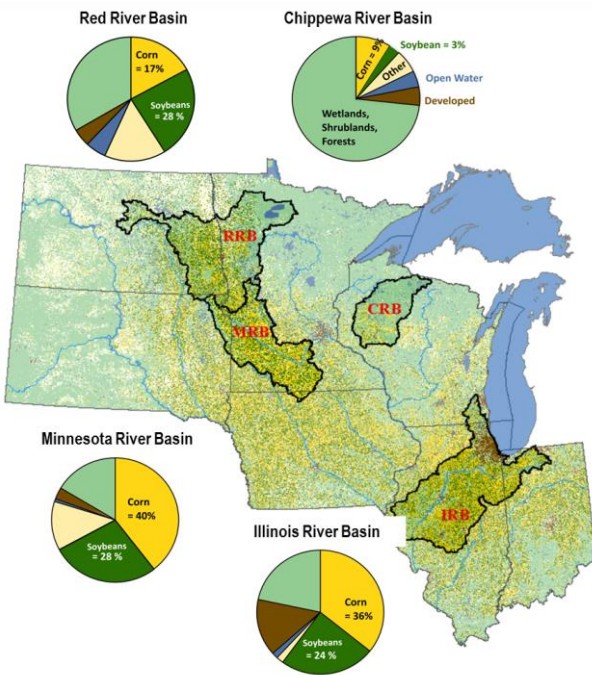

**Figure 1: 2013 Cropland Data Layer (CDL) showing the relative proportion each land cover class for the four study watersheds in the Upper Midwest.**

## 3 Data and Methods: land use land cover, climate, and streamflow

We explain our methods for addressing how land use, climate, and streamflows have changed during the 20th and 21st centuries in sections 3.1 thru 3.3. In section 3.4 we explain how the timing and timescales of prominent change were determined. We use a water budget to determine whether precipitation and evapotranspiration alone can explain runoff trends in section 3.5.

## 3.1 Records of land use land cover (LULC) change during the 20th and 21st centuries

We compiled county-level US Census of Agriculture drainage data from 1940, 1950, 1960, 1978, and 2012 for each study watershed, weighing partial counties by area (U.S. Bureau of the Census, 1942, 1952, 1961, 1981; U.S. Department of Agriculture, 2014a). Tabulations of drainage enterprises exclude lands draining less than 500 acres in all years except 1940



(U.S. Bureau of the Census, 1922, 1952). In 1940 and 2012, acres drained by ditches and tile were reported individually. To normalize the land area across basins of different sizes, we report the percentage of watershed area drained. While the uncertainties in these data are high, they are the best data available on a national scale. We use these data as a proxy for the relative drainage extent and expansion in each of the four large study basin, the smallest of which is still larger than 20

counties.

County level agricultural census drainage data are only available for five census years. Therefore, we also compiled annual USDA National Agricultural Statistics Service (NASS) crop acreage harvested in each basin following the methods of Foufoula-Georgiou et al. (2015). We report the percentage of corn, soybeans, and hay and small grains grown in each watershed from 1915 to 2015. Artificial drainage installation has typically coincided with the replacement of hay and small

grains for soybeans as shown in Fig. S1 in the Supplement (Burns, 1954; Hewes and Frandson, 1952). Therefore we use these annual crop data as another indication of LULC changes.

## 3.2 Climate records: precipitation and evapotranspiration

Monthly Parameter elevation Regression on Independent Slopes Model (PRISM) precipitation rasters produced by PRISM Climate Group (2004) and modeled actual evapotranspiration ($ET_a$) produced by Livneh et al. (2013) are readily

available, reproducible, and defensible climatology data that provide continuous spatial and temporal coverage of our study areas. We compiled spatially-averaged monthly and annual precipitation and evapotranspiration depths for each watershed for 1935-2013 and 1935-2011, respectively.

Livneh et al. (2013) evapotranspiration was produced for the continental United States using the Variable Infiltration Capacity (VIC) model run at 3-hr time steps in energy balance mode, consistent with methods of Maurer et al.

(2002). Hereafter we refer to Livneh et al. (2013) and Maurer et al. (2002) as L13 and M02. We have chosen L13 data over other available estimates of evapotranspiration because they cover a large spatial and temporal domain necessary for the study, i.e. the contiguous US from 1915-2011, at reasonable spatial ($1/16°$) and temporal (monthly) resolution, unlike other global and North American reanalysis products such as ERA-Interim (data available from 1979-2013 at $0.7°$) and NARR (data available from 1979-2015 at $0.3°$).

Although the precipitation input used to generate the $ET_a$ data was gridded NCDC COOP station data, Livneh et al. (2013) state that "gridded precipitation values were subsequently scaled on a monthly basis so as to match the long-term mean from the Parameter-Elevation Regressions on Independent Slopes Model (PRISM; Daly et al. 1994); for consistency with M02, a 1961–90 PRISM climatology was used". We directly compared monthly precipitation from L13 and PRISM (1935-2011) and found that for each of the four study basins the mean error was 1% (Fig. S3).





### 3.3 Streamflow gauge records

We evaluated annual (seasonal), monthly and daily flow metrics for each of the four river basins. Using multiple gauges for a single basin, we compiled seven annual flow metrics: mean annual flow, 7-day average annual low flow winter (November-April), 7-day average annual low flow summer (May-October), peak mean daily flow spring (March-May), peak mean daily flow summer and fall (June-November), high flow days, and extreme flow days using mean daily flow data from USGS gauges within each basin (Fig. S2; Table 1) following the methods of Novotny and Stefan (2007). The number of high and extreme flow days refers to the number of days in a given year that are one and two standard deviations above the 1950-2010 mean. For each gauge, we normalized the annual flow metric by the 1950-2010 mean to facilitate comparisons among basins and to observe similarities in trends among metrics. Each gauge record included a minimum of 62 years, and of the 63 gauges analysed 53 gauges had continuous records. Of the 10 non-continuous records, 4, 2, 2, 1, and 1 gauges were missing 2, 4, 6, 8, and 14 years of data respectively during the period 1929-2013 (Table 1).

For the downstream outlet gauge in each basin (Table 1) we computed annual and monthly streamflow average depths (cm month$^{-1}$) and volumes (km$^3$ month$^{-1}$) for 1935-2013 for the MRB, RRB, and CRB, and 1939-2013 for the IRB due to missing gauge data prior to 1939. We also calculated daily streamflow change exceedance probabilities, where dQ/dt>0 characterizes the rising limbs of daily hydrographs and dQ/dt<0 the falling limbs.

### 3.4 Determining the timing and time-scales of prominent LULC, climate, and runoff changes

In order to determine whether observed changes in climate and streamflow are statistically meaningful and potentially coincident with LULC change, we first determined the timing of climate, streamflow, and LULC change. Annual crop data reveal the timing of rapid expansion of soybean acreage and indicate land use land cover transitions (LCTs) when soybean acreage exceeds hay and small grains (Foufoula-Georgiou et al., 2015). We identified the timing of precipitation and streamflow change using wavelets and by fitting a piecewise linear regression (PwLR) using a least-squares approach to the monthly streamflow and precipitation volume time series in each basin (Liu et al., 2010; Tomé and Miranda, 2004; Verbesselt et al., 2010; Zeileis et al., 2003).

A common method for detecting and quantifying changes in the magnitude/frequency content of a time series is via a localized time-frequency analysis using wavelets. The Continuous Wavelet Transform (CWT) of a signal $x(t)$ is defined as the convolution of the signal with scaled and translated versions of a mother wavelet $\psi(t)$:

$$T(a,b) = \frac{1}{\sqrt{a}} \int_{-\infty}^{+\infty} x(t) \psi^* \left(\frac{t-b}{a}\right) dt \qquad (1)$$

where $\psi\left(\frac{t-b}{a}\right)$ is the mother wavelet scaled by parameter $a$ and translated by parameter $b$, and * denotes the complex conjugate. By changing $a$ (i.e., dilating or contracting the mother wavelet by a scale factor) and changing $b$ (centering the wavelet at different locations along the time axis), the CWT quantifies the localized energy or variance of a signal at





different times and scales (frequencies). To every scale there is a corresponding frequency assigned as the central frequency of the Fourier transform of the wavelet at that scale. This relationship is analytically computable depending on the chosen mother wavelet. In this paper, we use the Morlet wavelet (Addison, 2002; Daubechies, 1992; Seuront and Strutton, 2003), which has been proven effective for analyzing climate signals such as El Niño, streamflow, and precipitation among others

(e.g., Anctil and Coulibaly, 2004; Foufoula-Georgiou et al., 2015; Labat et al., 2001; Torrence and Compo, 1998 and the references therein). The Morlet wavelet is given as:

$$\psi(t) = \frac{1}{\pi^{1/4}}\left(e^{i2\pi f_0 t} - e^{-(2\pi f_0)^2/2}\right)e^{-t^2/2} \tag{2}$$

which is simply a complex wave with a Gaussian envelope, where $f_0$ is the central frequency of the mother wavelet. In practice, when $f_0 \gg 0$, the second term in the parenthesis of Eq. (2) becomes negligible and the Morlet wavelet simplifies to:

$$\psi(t) = \frac{1}{\pi^{1/4}}e^{i2\pi f_0 t}\,e^{-t^2/2}\,. \tag{3}$$

The Morlet wavelet with $f_0 = 0.849$ achieves the best compromise between time and frequency localizations.

We also evaluated precipitation and streamflow change using 1974/75 as a breakpoint for the pre-period and post-period because it lumps the time series data into two roughly equal periods (40/39 years), coincides with the timing of widespread acceptance of cheaper and easier to install corrugated plastic tile (Fouss and Reeve, 1987), and other studies in

the MRB and IRB have identified hydrologic change occurring around that time (e.g. Foufoula-Georgiou et al., 2015; Lian et al., 2012; Schottler et al., 2014). Acknowledging that 1974/75 may not be the hydrologically relevant breakpoint in all basins at this large scale, we ran statistical tests using 1974/1975 as well as the breakpoints identified for each basin from the PwLR and LCT.

We performed one-tailed student's t-tests or Wilcoxon Rank Sum tests when data did not meet parametric

assumptions after testing log, square root, and arcsine transformations, and Kolmogorov–Smirnov (KS) tests using the statistical program R to analyze changes in the mean and distribution of annual and monthly total flow (Q) at the basin outlet and spatially averaged basin precipitation (P) volumes between each pre-period and post-period (R Core Team, 2013). We test the hypothesis that mean monthly water volumes have increased and their distributions have shifted right during the post-period. We selected an alpha value of 0.05 (95% confidence level) for all statistical tests performed. Thus we performed

286 t-test and 268 KS-test using the annual and monthly P and Q data for each basin, as well as 28 t-tests on the seven streamflow metrics described in section 3.3 for a total of 600 statistical tests. In general the results of the statistical tests are not sensitive to the timing of different breakpoints, spanning nearly four decades, and therefore we generally report statistical results for the pre-period (1935-1974) and post-period (1975-2013), though all results are presented in Table S1 in the Supplement.





### 3.5 Determining the role of climate versus LULC change on streamflows using a water budget

For given watershed over a specified time period of integration, water inputs minus water outputs are equal to the change in storage per unit time:

$$P - ET - Q = \frac{dS}{dt} \qquad (3)$$

where $P$ is average watershed precipitation (cm month$^{-1}$), $ET$ is estimated average watershed actual evapotranspiration (cm month$^{-1}$), $Q$ is runoff depth at the basin outlet (cm month$^{-1}$), and $\frac{dS}{dt}$ is the depth of change in soil water, groundwater, and lake/reservoir storage per unit time.

We have computed average annual water budgets for each basin by accumulating monthly $P$, $ET$, and $Q$ during the pre-period and post-period determined by the land cover transition (LCT) and 1974/75 in each basin, to solve for the change in storage. If the change in storage term increases from the pre-period to post-period we conclude that soil moisture, groundwater, and/or lake/reservoir storage has also increased and that climate likely explains most of the increase in Q. However, if the change in storage term decreases from the pre-period to post-period, then we conclude that soil moisture, groundwater, and/or lake/reservoir storage has decreased despite precipitation increases, indicating that widespread LULC change has altered watershed storage and contributed, in addition to precipitation, to increased streamflows.

Livneh et al (2013) did not incorporate land use land cover changes, such as tile drainage expansion or crop changes, into the VIC model. The fact that LULC change is not included in the model is what allows us to test, external to the ET predictions, whether or not a LULC effect exists. There is no evidence of regional groundwater change and the effects of dams and urbanization on streamflows are likely minimal as discussed in section 2. Comparing these data to other estimates of evapotranspiration including four AmeriFlux towers, two of which are in corn-soy agricultural areas, we demonstrate that they are sufficiently reliable modern estimates for our purposes (Table 2; Fig. S4; Fig. S5).

We acknowledge that there is uncertainty in the all of the input data and understand that the magnitude of the storage term is sensitive to estimates of ET. Livneh et al. (2013) reported 17% overestimation of $ET_a$ during the summer months when compared with AmeriFlux station data. It is during summer months that ET is most likely limited by soil water availability. Therefore in addition to the raw water budgets, we present water budgets where we have reduced monthly $ET_a$ by 17% during summer months (JJA). This lower estimate of ET effectively reduces the potential amount of streamflow change that could be attributed to land use and artificial drainage and is therefore a more conservative analysis. Overall, the data from Livneh et al. (2013) used in computing the monthly water budgets are consistent with other sources (Bryan et al., 2015; Diak et al., 1998) and provide reasonable modern estimates of $ET_a$, especially when reducing summer (JJA) $ET_a$ by 17% (Figs. S4 & S5).



## 4 Results and Discussion

### 4.1 Drainage, corn and soybean expansion during the 20$^{th}$ and 21$^{st}$ centuries in the Upper Midwest

Across the Upper Midwest, the percent of land drained by tiles and ditches and cultivated for corn and soybeans has increased since the early twentieth century while land cultivated for hay and small grains has declined. Figure 2 shows the percent of each watershed drained by tiles and ditches from the Census of Agriculture data, as well as the percent of each county drained by tile in 1940 and 2012. Total drainage and tile drainage has increased in the MRB and IRB, while it has remained relatively unchanged from 1940 to 2012 in the CRB and RRB (Fig. 2). The drainage census data show that the MRB has the greatest percentage of the watershed area drained by tile, 19% in 1940 and 35% in 2012, and ditches, 7% in 1940 and 10% in 2012, followed closely by the IRB where tiles drained 10% and 34% of the watershed area in 1940 and 2012, respectively, and ditches drained 12% and 8% of the watershed area in 1940 and 2012, respectively (Fig. 2). The Red River of the North basin has experienced very little increase in total drainage since 1940. Most artificial drainage in the RRB is ditches rather than tile drains. Although a dramatic increase in tile installation has been reported in the Red River Valley since the 1990's, the area of this expansion appears small relative to the watershed area. Acres reported to be drained by tile in 2012 represents only 2% of the total watershed area. The Chippewa River Basin has very little agricultural land and thus the 2012 census reports less than 1.5% of the watershed area drained by tile and ditches (Fig. 2).

The 1978 census data illustrate the uncertainty associated with reporting and response rates, as it is unlikely for total drainage to have decreased between 1960 and 1978 in the RRB and MRB (Fig. 2). Most county ditches and tile in Blue Earth County, Minnesota were installed during the 1910's and 1920's with a noticeable drop off during WWII and a resurgence of drainage enterprises starting in the 1960's (Blue Earth County Minnesota, n.d.). Burns (1954) reported that the 1940 census data underestimated drainage enterprises in Blue Earth County by 8.5%, simply due to inaccuracies in reporting. Furthermore, in 2012, 82%, 80%, 51%, and 91% of all farms in Minnesota, Illinois, North Dakota, and Wisconsin, respectively, were less than 500 acres, and therefore were not included in survey results (U.S. Department of Agriculture, 2014b). Therefore these estimates are likely to underestimate the area drained by tile and ditches. Although the 2012 census attempts to correct for incomplete and missing responses, because drainage enterprise records have traditionally been so poorly documented, it is difficult to know how much reported acreage underestimates the actual acreage.

We also note that acres drained by tile and ditches do not directly translate to effectiveness of artificial drainage at transferring soil water to streams. Several factors influence the flow rate from soils to drains including the hydraulic conductivity of the soil, prevalence of soil macropores, depth of the water table, depths of the tile lines, diameter of the tile, slope of the tile or ditch, horizontal spacing of tiles and ditches, material composing the tile or ditch, as well as precipitation intensity and duration and antecedent soil conditions (Hillel, 1998). We simply do not have this level of information regarding artificial drainage in the Midwestern USA and suspect that the spatial variability in drainage management practices



may be high. For example, Naz et al. (2009) mapped tile drains in a 202 km$^2$ Indiana watershed and found tile spacing that ranged from 17-80 m.

While we expect that the drainage trends observed are relatively correct, we are cautious about drawing any definitive conclusions from the Census of Agriculture data regarding the actual extent of tile drainage and changes over time.

It is clear that these estimates tend to underestimate the amount of drainage. Nevertheless, total drainage and tile drainage in the Minnesota River basin and Illinois River basin have increased considerably since 1940. It is known anecdotally, but not included in these data, that tile drainage spacing has decreased and intensity or drainage rate in mm h$^{-1}$ has increased on agricultural lands, often by a factor of two, as was done at the Lamberton Research Station, MN (L. Klossner, personal communication, November 17, 2015). Overall we find artificial drainage extent greatest in the MRB and IRB, relatively low,

but growing recently in the RRB, and negligible in the CRB. While relative amounts of drainage in this inventory should be reliable, the lack of historical documentation on changes in location, density and type of tile installed limits our ability to model hydrologic change at the large landscape scale.

Conversion from small grains to soybeans is often accompanied by increased sub-surface drainage installation (Foufoula-Georgoiu et al., 2015). Figure 3 displays the percent of each basin harvested for corn, soybean, and hay and small

grains from 1915-2015. There has been a decline in hay and small grains and an increase in soybeans in all four of the watersheds over the period of record. The RRB is the only basin containing a significantly higher percentage of soybean acreage relative to corn; on average since 1995 soybean acreage in the RRB has been more than twice that of corn.

Overall, changes in crop type occurred gradually in the MRB and IRB, much more rapidly and recently in the RRB (Fig. 3). The CRB is largely non-agricultural, only 9% of the basin grew corn, soy, and hay and small grains in 2015, and the

changes in the basin have been small during the period of record (Fig. 3). While we cannot directly ascribe these changes in crop type to changes in drainage practices or vice versa, they provide a relatively detailed history of LULC and whether the changes occurred gradually or rapidly and recently or long-ago in each basin.





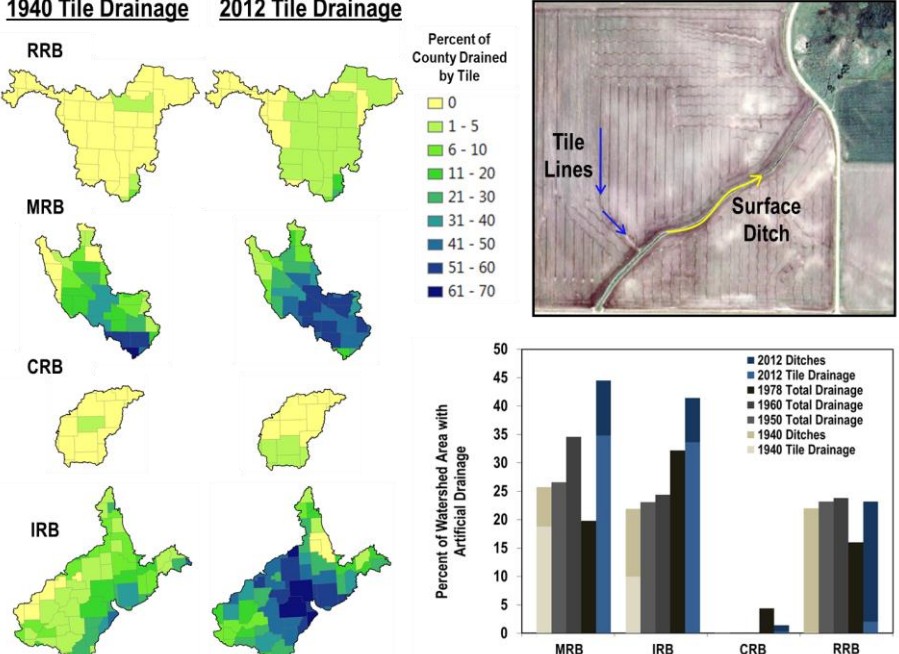

**Figure 2. At left: Spatial distribution of tile drainage patterns in each of the four study basins in 1940 and 2012. Upper right: image showing an example field pattern that combines subsurface tile lines with a surface ditch. Lower right: Percentage of the total watershed area with artificial drainage from 1940, 1950, 1960, 1978, and 2012 drainage census data. The magnitude of each bar indicates total drainage (ditches and tiles), and 1940 & 2012 bars are broken proportionally into drainage by ditches and tiles.**

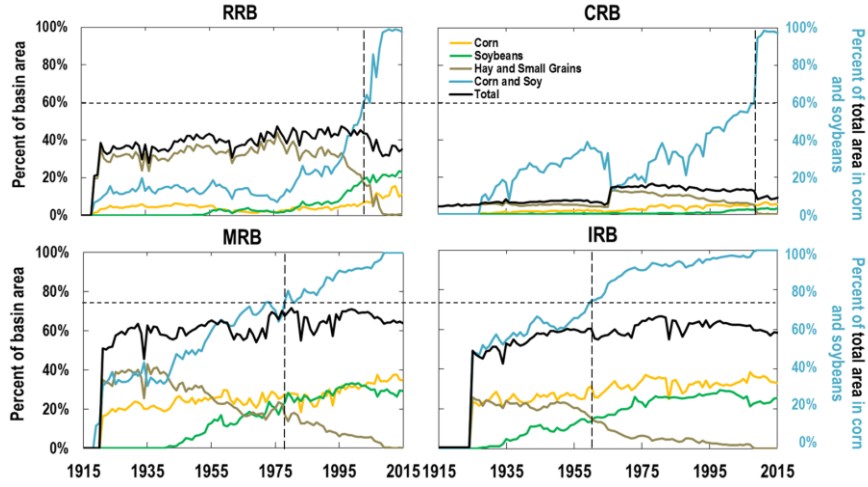

**Figure 3. Acres harvested of corn, soybeans, and hay and small grains (barley, oats, wheat) expressed as percent watershed area for each of the basins based on county level data from USDA NASS. The sum of these three commodity groups is shown as a total in black and the percent of this total area in corn and soybeans is plotted in blue. Vertical dashed lines indicate when percent of basin area harvested for soybeans exceeds hay and small grains. Horizontal dashed lines indicate when the percent of total area harvested for corn and soybeans exceeds 60% in the RRB and CRB and 75% in the MRB and IRB.**





### 4.2 Timing of streamflow change coincides more closely with precipitation than land cover change

The land cover transition (LCT), precipitation, and streamflow breakpoints of change identified using piecewise linear regression (PwLR) and continuous wavelet transform (CWT) reveal that the timing of precipitation and streamflow change generally preceded LCT change (Table 3). This was true for all tests in the RRB and CRB. However, there are some chronical differences in the order of precipitation, streamflow, and LCT breakpoints. In the IRB, the timing of LCT precedes precipitation and streamflow breakpoints identified using PwLR and CWT by between 13 years and 20 years (Table 3). In the MRB, LCT follows precipitation by 20 years and streamflow by 11 years as identified using PwLR but precedes the streamflow breakpoint by one year identified using CWT (Table 3).

Land cover transition breakpoints shown in Fig. 3 are not exact; land cover change occurs gradually, and therefore LCT breakpoints represent when a large portion of each watershed was converted to from hay and small grains to soybeans. Land cover transition breakpoints are indicated two ways: 1) when percent watershed area harvested for soybeans exceeds hay and small grains, and 2) when the proportion of the total acreage harvested for the three commodity groups is dominated by corn and soybeans. The second criteria varies from basin to basin, as some basins may have historically grown more hay and small grains, while others more corn and soybeans. In the CRB and RRB, hay and small grains exceeded 50% of the total area harvested for corn, soybeans, and hay and small grains from 1915 until the year 2000 or later. However in the MRB and IRB, hay and small grains only exceeded 50% of the total area harvested for the three commodity groups from 1915 until 1950 or earlier. The LCT breakpoints, indicated by the vertical dashed lines in Fig. 3, approximately coincide with the horizontal dashed lines, which represent a time when the percent of the total acres harvested for the three commodity groups exceeded 60% in RRB and CRB, where hay and small grains have historically dominated, and 75% in the MRB and IRB, where corn and soybeans have historically dominated. We acknowledge that these breakpoints do not consider the actual extent of soybeans, which is assumed to be a surrogate approximation for area of drained croplands. Soybean coverage is much higher for both MRB and IRB compared to RRB and CRB even before 1955. Considering the large proportion of the MRB and IRB watersheds cultivated for soybeans in the early 1950's combined with extensive (20-25%) drainage by 1940 and 1950 (Fig. 2), this suggests streamflow changes generally occurred after both precipitation and LCT changes.

We observe minimal changes in the energy of the annual and inter-annual precipitation signal for any basins during the period of record, and therefore could not identify the timing of precipitation change in any basin using CWT (Fig. 4). However, Figure 4 displays significant increases in the annual and inter-annual energy of the basin outlet streamflow signal around 1975, 1980, and 1995 for the IRB, MRB, and RRB respectively, while the CRB does not exhibit any striking changes in energy throughout the period of record. All decadal energy shifts in the precipitation signals are clearly translated into the decadal energy of the streamflow signals for all four basins (Fig. 4). The observed correlation between the decadal energy changes in streamflow and precipitation signals together with the lack of any significant correlation between their energies at

the annual scale may signal the importance of factors other than precipitation, here artificial drainage, to streamflows in the MRB, RRB, and IRB at the annual scale.

In all basins, the timing of precipitation change coincided with or preceded streamflow breakpoints based on PwLR (Table 3). Similar temporal coincidence of precipitation and streamflow breakpoints in contrast to the LCT and streamflow breakpoints may suggest that streamflow changes are tightly coupled with precipitation changes. However, that interpretation fails to account for the potential effects of drainage, which could amplify the streamflow response to precipitation.

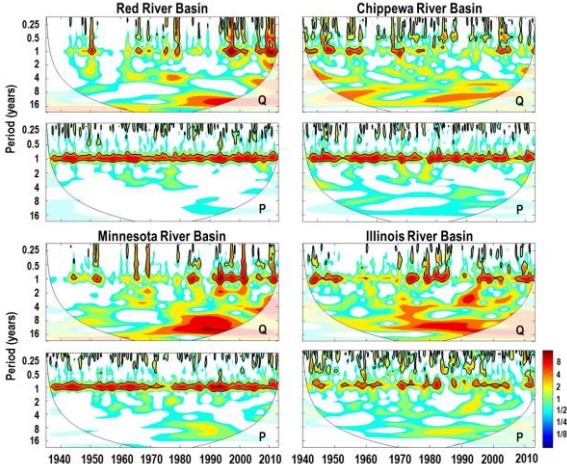

**Figure 4. Continuous Wavelet Transform (CWT) energies for monthly volumetric streamflow (Q) and precipitation (P) time series.**

### 4.2.1 Seasonal and annual scale changes of precipitation, evapotranspiration, and streamflow

The raw timeseries of spatially averaged annual precipitation and streamflow depths (cm), reported in the Supplement, show an increasing trend in precipitation and streamflow in the RRB, MRB, and IRB and no trend in the CRB (Fig. S6). The magnitude of the precipitation and streamflow trends are on the order of 120-150 mm/century and 90-170 mm/century, respectively, and are consistent with those reported for the entire Upper Mississippi River basin by Frans et al. (2013). Figure 5a shows five year running averages of seven annual streamflow metrics, where normalized values of 1 indicate that the annual value is equivalent to the mean (1950-2010) value. Stationary flow statistics vary around 1 for the entire time series, as is the case for the Chippewa River basin (Fig. 5). Non-stationary time series systematically deviate from 1, indicating that the mean condition has changed during the period of record. Qualitatively, all seven flow metrics in the CRB have remained stable since the 1930's, except for seven day low flows in winter, which have increased 12% since 1975 (p<0.01) (Fig. 5).



Unlike the Chippewa, flow metrics in the Minnesota, Red, and Illinois river basins systematically increase in recent decades, with nearly a two-fold increase or greater in almost all flow metrics since 1975 (Fig. 5). Seven day low flows in summer and winter have increased most in these basins; means have increased 67%-275% (p<0.001) since 1975 (Fig. 5b). High flow and extreme flow days have also increased significantly in the MRB (p<0.001), IRB (p<0.05) and RRB (p<0.001). Spring peak daily flows have changed the least in all basins, indicating 14% (p>0.05), 37% (p<0.05), and 60% (p<0.05) increase in mean between 1934-1974 and 1975-2013 for the IRB, MRB, and RRB, respectively (Fig. 5b). The Minnesota River basin has seen the greatest percent increase in mean annual flow, peak daily flow summer & fall, 7 day low flow in winter, high flow days and extreme flow days (Fig 5b). Peak daily flow summer and 7 day low flow in summer have increased most in the Red River of the North basin (Fig. 5b).

All seven flow statistics in the Red River of the North basin increase dramatically after the mid-1990's (Fig. 5a). Low flows have increased 3.5-4 fold (p<0.001) and high and extreme flows have increased 2.5-3 fold (p<0.001) in the RRB since 1995 (Fig. 5b). Flows in Minnesota River basin have increased similarly, with a 3-4 fold increase in low flows (p<0.001) and 3 fold increase in high and extreme flows (p<0.001) since the timing of land cover transition. Changes in the Illinois River basin are less obvious, yet still significant, with a 2 fold increase in low flows (p<0.001) and 1.5 fold increase in high and extreme flows (p<0.05) since LCT.

The MRB and RRB exhibit an increase not only in the magnitude but also in the cyclicity and synchronicity of these metrics after about 1980 (Fig. 5a). Cyclicity could imply that climate is playing a role in the observed increase in flows. However, the extent to which agricultural land and water management practices may be amplifying this climate effect cannot be ascertained from this figure alone. The Illinois River basin exhibits the most change in summer and winter 7 day low flows, which increase after 1970, and this trend is even more pronounced when only examining gauges within predominantly agricultural sub-basins that are unaffected by large dams (Fig. 5a). However, the changes in the RRB and MRB are much more obvious and statistically significant than those in the IRB.

Statistical results for annual changes in streamflow and precipitation for all breakpoints can be found in Table S1 in the Supplement. The following results are based on the 1974/75 breakpoint. Overall, average annual streamflow, precipitation, and evapotranspiration depths have increased significantly in the MRB and RRB, while only streamflow has increased significantly in the IRB; no significant changes are reported in the CRB. Average annual runoff depth at the outlet gauge of the MRB has increased 5.9 cm (p<0.001). Average annual precipitation and evapotranspiration depths in the MRB have also increased by 4.6 cm (p=0.033) and 3.3 cm (p=0.021), respectively. Average annual runoff ratio has increased from 0.11 to 0.18, equivalent to a 65% increase and consistent with the results of Vandegrift and Stefan (2010). In the RRB, the average annual runoff ratio has increased 65%, from 0.07 to 0.11 at the outlet gauge, which is slightly greater than the 55% increase reported by Vandegrift and Stefan (2010). On average, annual runoff, precipitation, and evapotranspiration depths have increased by 2.9 cm (p<0.01), 4.1 cm (p=0.019), and 2.4 cm (p=0.043), respectively. Average annual runoff in the IRB



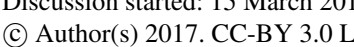
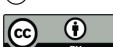

has increased 5.4 cm (p=0.011). Precipitation and evapotranspiration are likely increasing in the IRB, however given the statistical power the apparent 4.2 cm (p=0.086) and 1.9 cm (p=0.072) increases were not significant. The average annual runoff ratio in the IRB has increased from 0.30 to 0.34, a 14% increase. The CRB average runoff ratio has decreased slightly (2%), from 0.37 to 0.36. On average, annual runoff depth in the CRB has not changed (0.00 cm; p=0.499). Average

5 precipitation and evapotranspiration depths may have increased slightly, perhaps as much as 2.0 cm (p=0.243) and 0.9 cm (p=0.209) respectively, but these changes were not statistically significant.

The MRB and RRB exhibit the greatest change in the annual runoff ratio, followed by the IRB, with negligible change in the CRB. These findings are consistent with the fact that the MRB and RRB have relatively low runoff ratios comparted to the CRB and IRB, and are the only two basins where annual precipitation and evapotranspiration increases

10 were statistically significant. On average, the fraction of annual precipitation that goes as ET has decreased 1.0%-2.4% in all four study basins, which is smaller in magnitude but consistent in direction of change with Schottler et al. (2014) who found the ratio of PET/P decreased 5.6% between 1940-1974 and 1975-2009 in a subbasin of the MRB. Schottler et al. (2014) considered the effects of both climate and cropping practices in calculations of PET while the Livneh et al. (2013) calculated $ET_a$ only considering climate.

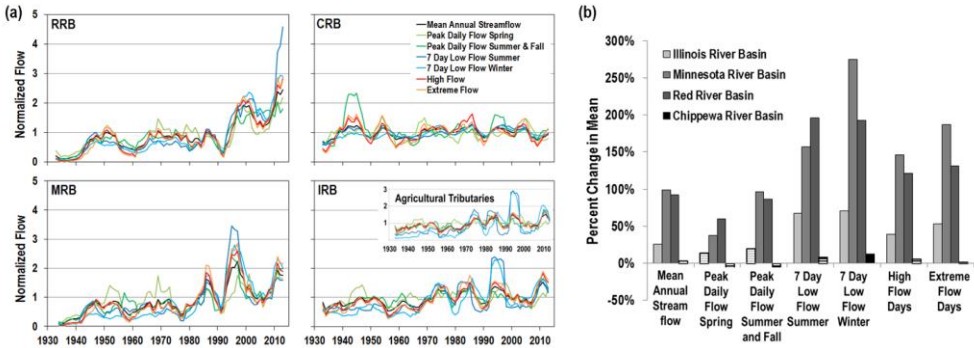

**Figure 5. a) Seven normalized streamflow statistics presented as five year running averages based on annual and daily gauge analysis for the Red River of the North (RRB - 22 gauge), Chippewa River (CRB - 9 gauges), Minnesota River (MRB - 12 gauges), and Illinois River (IRB - 20 gauges); IRB inset of 7 tributary gauges, predominately agricultural, not influenced by major dams. b) Percent change in flow metric mean between 1934-1974 and 1975-2013. Solid bars indicate significant increases in means (alpha=0.05).**

### 4.2.2 Monthly scale changes of precipitation, evapotranspiration, and streamflow

Cumulative monthly precipitation, plotted in Fig. 6, indicates no systematic change in cumulative precipitation with time (i.e. constant slope) for any basin. However, cumulative monthly streamflow (1935-2013) plotted in Fig. 6 indicates a sudden change in slope around 1973 in the IRB, 1980 in the MRB, and 1995 in the RRB, without a distinct change in slope

25 in the CRB. The visually identified change points are consistent with those identified from the CWT (Fig. 4).



Statistical tests of monthly streamflow and precipitation resulted in the same interpretations for 95% of the tests regardless of the breakpoint (Table S1); therefore Fig. 7 summarizes the results of these statistical tests for flow and precipitation in all basins using the 1974/75 breakpoint. Figure 7a illustrates the kernel density estimation, or non-parametric estimation of the probability density function, during the pre-period and post-period for June and September flows in each

basin. Figure 7b reports 192 results (48 p-values reported per basin) from the monthly streamflow and precipitation t-tests and KS tests. Each color wheel displays 24 results, 2 results per month for each basin, and shows significant p-values for t-tests and KS tests based on color. Color is inversely related to p-value such that smaller p-values and thus more significant results are shown in increasingly darker colors, with p-values greater than 0.05 colored white. As such the streamflow color wheel in Fig. 7b for the Chippewa River basin is completely white, indicating there were no statistically significant changes

in the mean or distribution of monthly streamflow volumes for any months, consistent with the assessment of the seven annual streamflow metrics and cumulative streamflow (Figs 5 and 6). We report a significant increase in mean October precipitation in the CRB. Monthly results for flow and precipitation changes in the CRB are consistent with the annual changes reported earlier.

In stark contrast to the CRB, the streamflow color wheels for the MRB and RRB show significant changes in mean

or distribution of monthly streamflow for nearly all months (22 out of 24 for MRB and 21 out of 24 for RRB) (Fig. 7b). In the RRB, mean precipitation in October has increased, and the precipitation distributions have shifted to the right for September and October (Fig. 7b). In the MRB, there has been a significant increase in mean March precipitation (Fig. 7b). The IRB exhibits fewer overall changes in streamflow than the RRB and MRB, with significant changes in monthly streamflow volumes for September, October, November, December and March, and significant changes in August and

November precipitation (Fig. 7b).

We acknowledge that due to high variability and small sample sizes, we may not have sufficient power to detect small, but real changes in precipitation and streamflow using these statistical tests, and thus may be prone to Type II error (Belmont et al., 2016). However, these results are consistent with the qualitative assessment of CWT, results of the seven annual flow statistics, and cumulative precipitation and streamflow trends, which indicate only slight changes in total

precipitation across all basins, large increases in total flow in the MRB and RRB, moderate flow increases in the IRB, and no streamflow changes in the CRB (Figs 4, 5, and 6).

To understand whether the cause and effect interconnection of streamflow (Q) and precipitation (P) has changed we plotted the joint probability distribution functions (joint PDF) of monthly P and Q, $f(P, Q)$, for each basin (Fig. 8). Joint PDF of pairs of monthly P and Q is the chance of their occurrence simultaneously. In Fig. 8 we illustrate three empirical quantiles

of the joint PDFs through contour levels $\alpha \in \{0.1, 0.6, 0.9\}$, where each contour level represents the boundary of a discrete 2D space in which the probability of each (P, Q) pair to fall inside that 2D space is alpha. A shift in the contour levels in the vertical, rather than diagonal, direction suggests that changes in precipitation magnitude alone cannot explain changes in





streamflow, and some other component of the system must be amplifying the transformation of precipitation to runoff at the monthly timescale.

There is a shift toward larger monthly streamflow volume for the same volume of precipitation at each 10% and 60% quantile in the MRB and 60% and 90% quantile in the RRB (Fig. 8). However it appears the 90% exceedance contour
5 for the MRB and 10% exceedance contour for the RRB have shifted up and to the right, indicating that an increase in precipitation in the driest months in the MRB and wettest months in the RRB could also be driving some of the change in flow (Fig. 8). Certainly the largest observable change in the MRB and RRB during this time is a shift from small grains to soybeans and an increase in the density and efficiency of drain tile networks. While analyses shown above documented significant changes in streamflow of IRB (Figs. 4, 5, 6, and 7b), this change is not as obvious in these joint PDF contours,
10 which indicate only a slight vertical shift in all quantiles (Fig. 8). Consistent with other analyses, the CRB does not demonstrate any shift in the P-Q relation suggesting the streamflow has been largely unaffected by the observed slight increase in annual precipitation in the basin (Fig. 8).

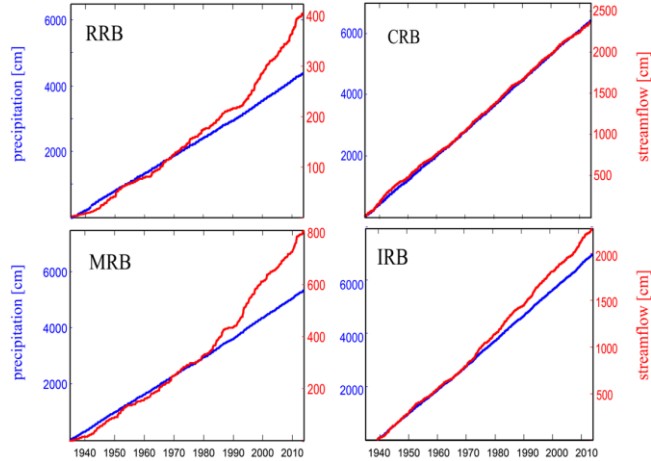

15 **Figure 6. Cumulative monthly precipitation (blue) and streamflow (red) depths (cm) for each river basin. Breakpoints, where the streamflow-precipitation relationship starts to change, are hard to detect from the time series alone but can be clearly seen from the cumulative plots of the monthly data (i.e., when similar increments of monthly precipitation are translated into larger amounts of monthly streamflow).**



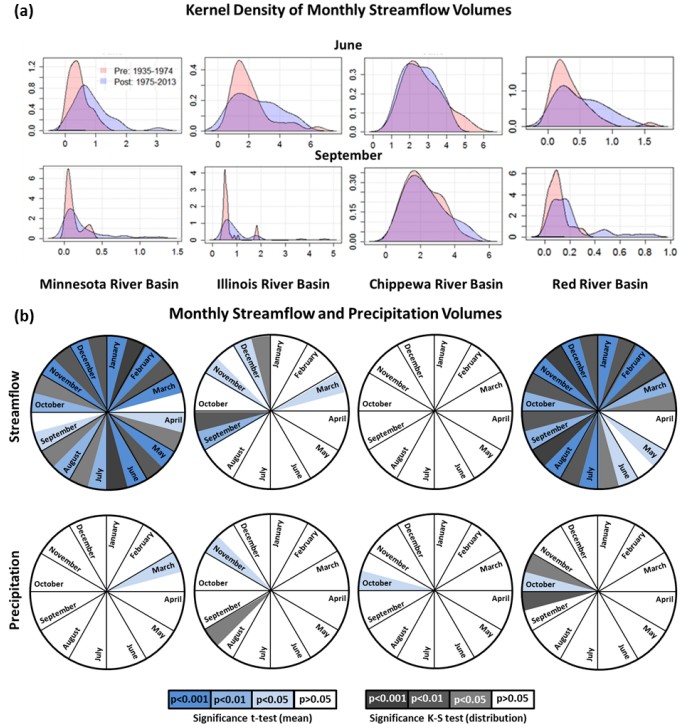

**Figure 7. a) Kernel density plots of monthly streamflow volumes for June and September for each basin b) Corresponding significance results for t-tests and K-S tests (α=0.05) of monthly streamflow and precipitation volumes, where a significant result indicates a positive shift (increase) in the mean or distribution between 1935-1974 and 1975-2013.**

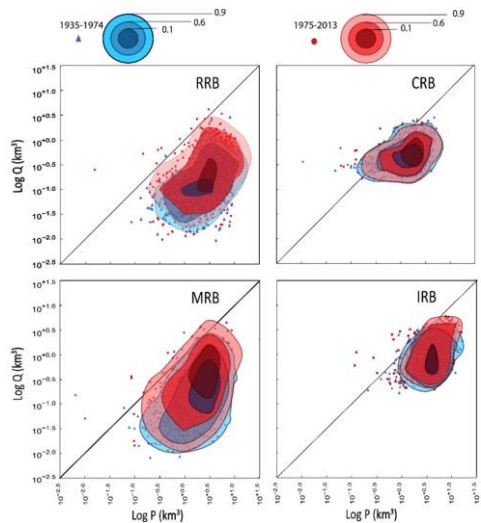

**Figure 8. Log-log empirical quantiles of joint PDF plots of monthly streamflow (Q) versus monthly precipitation (P) volumes for each river basin during the pre-period (blue: 1935-1974) and post-period (red: 1975-2013); bulls eye shading represent the 0.1 (dark), 0.6 (medium), and 0.9 (light) confidence intervals.**





### 4.2.3 Daily scale changes of streamflow

At the daily scale, we found an increase in the magnitude of streamflow change (hydrograph slopes) for both the daily rising limbs (dQ/dt>0) and falling limbs (dQ/dt<0) of the hydrographs for RRB, MRB, and IRB outlet gauges, suggesting an increase in "flashiness", or daily rate of change, of the hydrologic response (Fig. 9). Although the greatest average daily rates of change are less in the post-period than pre-period in the MRB for probabilities of exceedance less than 0.2%, this constitutes a small fraction of the total observations and can be linked to extreme events (Fig. 9). Figure 9 shows a slight decrease in the post-period curve for the CRB, indicating that the rising limb and falling limb flows may actually be less "flashy" in recent times than in the past. May-June is approximately the start of the growing season for soybean and corn and it is the time that tiles are most active, as this time of year usually corresponds to high monthly rainfall, high antecedent moisture conditions from spring snowmelt, and lower ET rates than the peak growing season due to lower crop water demands, and air temperatures that precede the annual peak. Considering rising and falling limbs exclusively in May-June, the magnitude of changes are even greater than the "all months" period for the MRB and RRB (Fig. 9). May-June "flashiness" decreased for the same time period in the CRB.





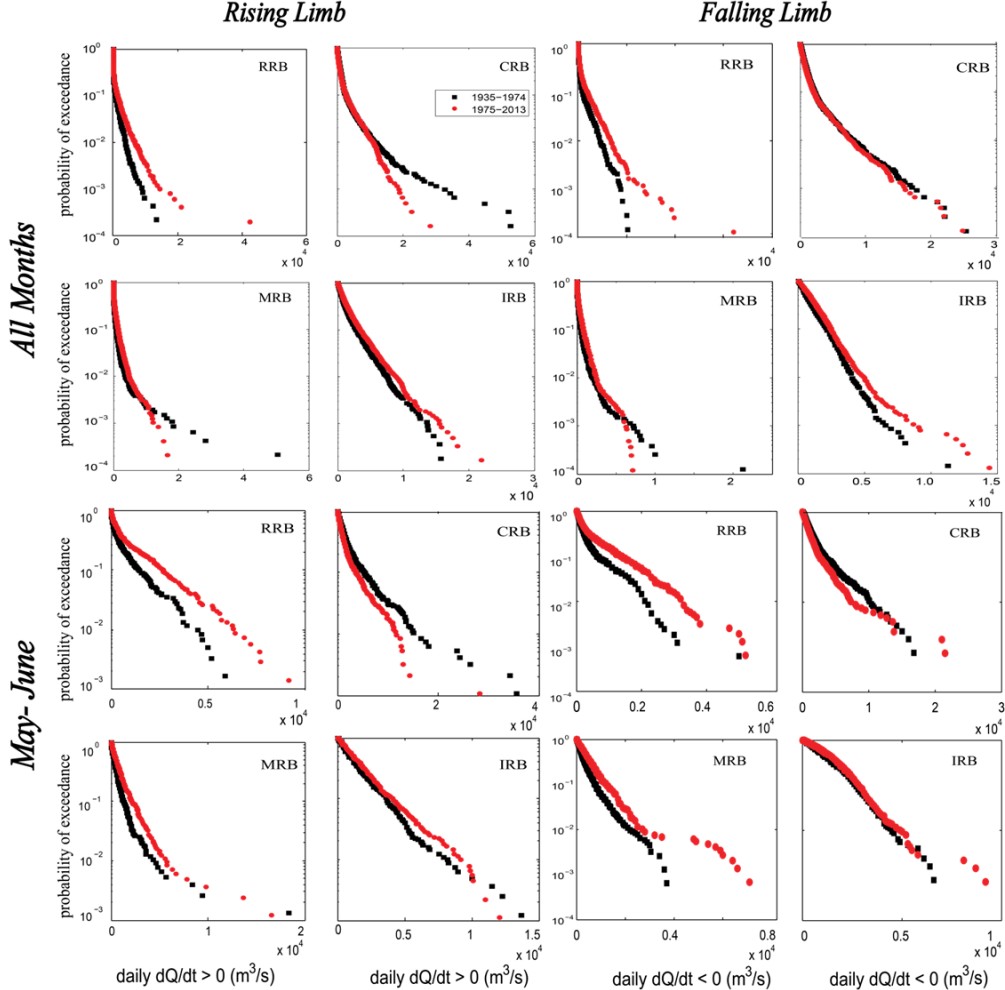

**Figure 9. Daily streamflow change exceedance probabilities, where daily (dQ/dt>0) characterizes rising limb flows and daily (dQ/dt<0) characterizes falling limb flows. Exceedance probability is computed for all months of the year (top two rows) and only May-June (bottom two rows) separately.**

**4.3 Hydrologic budgets suggest declining watershed storage in agricultural basins**

While time series and statistical analyses reveal useful insights regarding the timing, magnitude, and significance of precipitation and streamflow changes, as well as provide a qualitative indication of whether or not changes in precipitation and streamflow may be correlated and proportional, they cannot fully deconvolve or attribute the influence of artificial drainage and climate on streamflows (Harrigan et al., 2014). Therefore, we calculate water budgets for each basin as a tool to



understand whether the observed changes in precipitation are large enough to account for the changes in streamflow, and if there is more or less watershed storage in recent times than in the past (Healy et al., 2007).

Table 4 reports the calculated average annual water budget terms – precipitation, streamflow, evapotranspiration, and change in storage – during the periods before and after the 1974/1975 and LCT breakpoint using raw and conservative (reduced by 17% in JJA) estimates of $ET_a$. We find that regardless of the breakpoint or raw vs. conservative estimates of $ET_a$ there is a net reduction in water stored in soil, groundwater, and/or lakes, wetlands, or reservoirs between the pre period and post period in the MRB, RRB, and IRB (Table 4). The most parsimonious explanation for this reduction in water storage is the systematic removal of wetlands and lowering the water table, accomplished through tile drainage installation and expansion.

The CRB, which is not intensively drained (Fig. 2) and has experienced little change in crop type (Fig. 3), has been subject to an increase in precipitation, but does not exhibit an increase in runoff (Table 4), consistent with Figs. 8 & 9b. The overall trends in the CRB water budget indicate that water storage may have actually increased slightly between the pre-period and post-period, which could be accomplished through increased soil moisture, groundwater recharge, or reservoir storage in recent times.

Using conservative estimates of summer $ET_a$ the change in storage term has decreased by about 200%, 100%, and 30%, in the MRB, IRB, and RRB from the pre-LCT-period to post-LCT-period. In the CRB, change in storage has increased by roughly 30% from 1935-1974 to 1975-2011. These results are consistent with our hypothesis that increases in artificial drainage in the MRB, RRB, and IRB necessarily change how precipitation is transformed into streamflow and that increases in precipitation alone cannot explain changes in streamflow in these basins. Without pervasive artificial drainage in the CRB, while precipitation has increased slightly, flows have not changed, likely due to increases in soil moisture, groundwater, and/or lake, wetland and reservoir storage. Seasonal changes in storage shown in Fig. 10 suggest that soil moisture, groundwater, and/or lake, wetland, and reservoir storage in the spring and summer is negative, suggesting not enough P given $ET_a$ to produce observed flows, and positive in the fall suggesting more P and $ET_a$ than necessary to produce observed flows and thus an increase in storage during the fall.

The Red River of the North and Minnesota River basins have some of the poorest drained soils of the Upper Midwest and historically grew more hay and small grains than the other basins (Fig. 3). The introduction of artificial drainage combined with the replacement of hay and small grains with soybeans and the lack of major dams and municipal and industrial water use, has resulted in pronounced streamflow amplification in response to land use and climate changes in the RRB and MRB relative to the IRB and CRB (Fig. 4). Additionally these two basins have seen greater changes in annual and even monthly precipitation (Figs. 7 and 8). However, the extensively drained Minnesota River Basin has seen the largest increases in flow and largest decrease in watershed storage for relatively similar climatic change to the IRB and RRB, and





this is likely because of the high degree of watershed hydrologic alteration and connectivity from drainage and lack of other anthropogenic water uses.

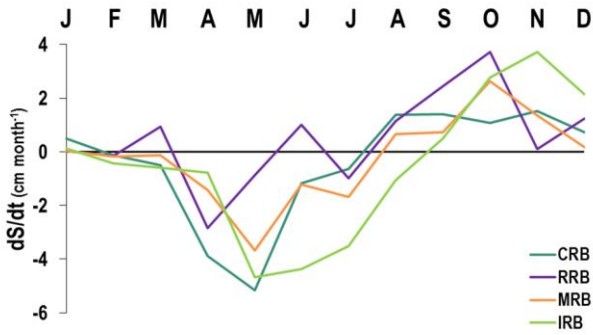

**Figure 10. Average monthly ds/dt calculated after LCT for IRB, MRB, and RRB, and after 1975 for CRB assuming 17% reduction in $ET_a$ for summer months.**

## 5 Interpretations, implications, and conclusions

The combined results of this study lead us to three conclusions regarding artificial drainage, climate, and streamflow change in the Upper Midwest during the 20[th] and 21[st] centuries: 1) widespread drainage expansion and intensification, especially of tile drainage, coupled with conversion of hay and small grains to corn and soybeans is evident and continues to occur in agricultural river basins; 2) annual precipitation and evapotranspiration have increased since 1975, though we found these changes to only be statistically significant in the MRB and RRB; monthly precipitation increases are generally not significant except in fall months; 3) across multiple scales (daily, monthly , annual) and for a range of flows (low, mean, extreme) streamflows have increased at all times of year in intensively managed agricultural watersheds (IRB, MRB, and RRB) and have remained stationary in the more forested CRB. The magnitude and timing of precipitation increases in each watershed suggests that precipitation strongly contributes to recently observed increases in streamflow, and agrees with the findings in the Upper Mississippi River basin (UMRB) (Frans et al., 2013). Despite this apparent correlation, the magnitude of precipitation increases alone cannot explain the observed increases in flow for agricultural basins according to the water balances. Therefore, it appears that the pervasive and extensive artificial drainage in agricultural basins has contributed to increased streamflow, not only at $10^2$-$10^3$ km watershed scales (e.g. Foufoula-Georgiou et al., 2015; Harrigan et al., 2014; Schilling and Libra, 2003; Schottler et al., 2014; Zhang and Schilling, 2006), but also at the scale of very large basins studied here.

Harrigan et al. (2014) recognize that often multiple drivers explain hydrologic change. These drivers are not mutually exclusive and may even act synergistically to explain observed streamflow trends. In the Midwestern USA possible explanations that could explain substantial streamflow increases include: 1) changes in storm duration and intensity or the



amount of precipitation falling as rain versus snow, have changed the characteristics of runoff generation while having little change on monthly or annual precipitation magnitudes; 2) increases in precipitation have translated into increases in soil moisture, which contributes to amplified flows; and 3) artificial drainage more efficiently routes sub-surface flow to streams, an effect which could be amplified by increased precipitation.

5       First, it is theoretically possible to observe changes in streamflow while having no change in monthly or annual precipitation magnitudes. High intensity, short duration events yield higher runoff ratios in poorly drained soils. Additionally warmer winter temperatures, earlier snowmelt, and more days when winter precipitation falls as rain instead of snow should affect and even increase winter baseflows, decrease the timing of ice break-up, and affect the magnitude of snowmelt floods. Several studies have documented such hydroclimate changes in the Midwestern USA (Feng and Hu, 2007; Groisman et al.,

2001; Higgins and Kousky, 2012) and the role of these hydroclimate changes could be explored by future investigations.

      Second, increased soil moisture is known to cause a nonlinear increase in runoff generation for similar precipitation events. Meyles et al. (2003) and Penna et al. (2011) report a threshold response in runoff generation when antecedent soil moisture exceeds ~65% of the soil porosity. It is possible that soil moisture has increased throughout the Midwestern US. However, no theory exists to predict how big this effect could be on landscape scales. Furthermore, there are very limited

data to determine whether or not soil moisture has in fact increased beyond such a threshold despite the immense amount of additional tile drainage that has been installed in the past few decades. Investigating this effect would be a good future step in this line of research.

      Third, several previous studies have demonstrated that artificial drainage increases streamflow in moderate sized ($10^2$-$10^3$ km$^2$) watersheds (Schottler et al., 2014; Foufoula-Georgiou et al., 2015). Though we cannot fully rule out the first

and second mechanisms discussed above, artificial drainage for corn-soy agriculture affects substantial swaths of land in all study watersheds except the Chippewa, and has almost doubled in area in the MRB and IRB since 1940 according to the US Census of Agriculture reports. It is known anecdotally that drainage has increased in density and efficiency during this same time. Using multiple lines of evidence from the analyses of very large basins and sub-basins it appears most likely that widespread agricultural drainage activities have amplified the streamflow response to relatively small changes in total

precipitation. Frans et al. (2013) found that artificial drainage amplified annual runoff in the UMRB in some cases by as much as 40% locally. Improved information regarding the size, spacing, depth, and extent of artificial drainage would greatly enhance our ability to model agricultural systems and predict downstream impacts.

      Surface and subsurface drainage remains an economically beneficial, yet largely unregulated land management practice in the Midwestern USA and Canada that affects enormous swaths of agricultural land (Cortus et al., 2011). Drainage

census data are prone to reporting inconsistencies and errors, overall underestimation of drainage from excluding farms less than 500 acres, and do not provide the information necessary for modeling basin hydrology in large agricultural watersheds (such as drain size, depth, spacing, and extent). However, these are the most comprehensive inventory of drainage in the

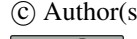



United States. This raises the question: why is such a widespread practice with such potentially profound and pervasive impacts on watershed hydrology and water quality so poorly documented and regulated? Until we have the information necessary to calibrate and validate watershed models, it will be difficult to more precisely deconvolve proportional impacts of climate and artificial drainage on flows at large spatial scales.

Though artificial drainage reduces field erosion by reducing surface runoff, it has been shown to essentially shift the sediment source from fields to channels (Belmont et al., 2011). The cause of that shift is still linked to agricultural land use. Basins experiencing increases in streamflow due to natural (climate) and anthropogenic (drainage) factors have increased stream power and are expected therefore to erode and transport more sediments and sediment bound nutrients and contaminants. Future increases in precipitation are likely to further intensify the effects of artificial drainage. Runoff

management, specifically increased residence time and damped peak flows, is most needed in spring and early summer when tiles are actively draining soils and precipitation events are large. Thus, substantial gains in water quality might only be achieved if some amount of the lost water storage capacity is reintroduced (e.g., wetlands, detention basins) into these agricultural watersheds.

### Acknowledgements

This material is based upon work supported by the National Science Foundation (Grant No. EAR-1209402) under the Water Sustainability and Climate Program (WSC): REACH (REsilience under Accelerated CHange), and by the National Science Foundation Graduate Research Fellowship Program under Grant No. 1147384. This research was supported by the Utah Agricultural Experiment Station, Utah State University, and approved as journal paper number 8938. The authors would like to thank Jon Czuba at Indiana University for his contributions towards the watershed crop analysis, and Karthik

Kumarasamy, Eden Furtak-Cole, and Mitchell Donovan at Utah State University for their input. CPC soil moisture data and Livneh et al. 2013 evapotranspiration data provided by the NOAA/OAR/ESRL PSD, Boulder, Colorado, USA, from their website at http://www.esrl.noaa.gov/psd/. Thank you to Alexander Bryan at the University of Michigan for generously providing evapotranspiration data from Bryan et al. 2015. Funding for AmeriFlux data resources was provided by the U.S. Department of Energy's Office of Science.

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

**Table 1. USGS stream gauging stations listed by study basin.**

| USGS gaging station | Station name | Period of record | Length (years) | Notes |
|---|---|---|---|---|
| *Chippewa River Basin (9 gauges)* | | | | |
| 05356000 | Chippewa River at Bishops Bridge, near Winter, WI | 1929-2013 | 85 | Mainstem river |
| 05356500 | Chippewa River near Bruce, WI | 1929-2013 | 85 | Mainstem river |
| 05360500 | Flambeau River near Bruce, WI | 1952-2013 | 62 | |
| 05362000 | Jump River at Sheldon, WI | 1929-2013 | 85 | |
| 05365500 | Chippewa River at Chippewa Falls, WI | 1929-2013 | 81 | Missing data: 1983 - 1986 |
| 05369000 | Red Cedar River at Menomine, WI | 1929-2013 | 85 | |
| 05368000 | Hay River at Wheeler, WI | 1951-2013 | 63 | |
| 05370000 | Eau Galle River at Spring Valley, WI | 1945-2013 | 69 | |
| 05369500 | Chippewa River at Durand, WI | 1929-2013 | 85 | Mainstem river - Downstream gauge |
| *Illinois River Basin (20 gauges)* | | | | |
| 05552500 | Fox River at Dayton, IL | 1929-2013 | 85 | |
| 05543500 | Illinois River at Marseilles, IL | 1929-2013 | 85 | Mainstem river |
| 05555300 | Vermilion River near Leonore, IL | 1932-2013 | 82 | † |
| 05556500 | Big Bureau Creek at Princeton, IL | 1937-2013 | 77 | † |
| 05554500 | Vermilion River at Pontiac, IL | 1943-2013 | 71 | † |
| 05569500 | Spoon River at London Mills, IL | 1943-2013 | 71 | |
| 05567500 | Mackinaw River near Congerville, IL | 1945-2013 | 69 | † |
| 05568500 | Illinois River at Kingston Mines, IL | 1940-2013 | 74 | Mainstem river |





| 05570000 | Spoon River at Seville, IL | 1929-2013 | 85 | |
|---|---|---|---|---|
| 05584500 | La Moine River at Colmar, IL | 1945-2013 | 69 | |
| 05585000 | La Moine River at Ripley, IL | 1929-2013 | 85 | |
| 05583000 | Sangamon River Near Oakford, IL | 1929-2013 | 79 | Missing data: 1934 - 1939 |
| 05582000 | Salt Creek near Greenview, IL | 1942-2013 | 72 | |
| 05580000 | Kickapoo Creek at Waynesville, IL | 1948-2013 | 66 | † * |
| 05578500 | Salt Creek near Rowell, IL | 1943-2013 | 71 | |
| 05572000 | Sangamon River at Monticello, IL | 1929-2013 | 85 | † |
| 05576000 | South Fork Sangamon River near Rochester, IL | 1950-2013 | 64 | |
| 05577500 | Spring Creek at Springfield, IL | 1949-2013 | 65 | † |
| 05586100 | Illinois River at Valley City, IL | 1939-2013 | 75 | Mainstem river - Downstream gauge |
| 05587000 | Macoupin Creek near Kane, IL | 1929-2013 | 77 | Missing data: 1933 - 1940 |
| *Minnesota River Basin (12 gauges)* | | | | |
| 05291000 | Whetstone River near Big Stone City, SD | 1932–2013 | 82 | |
| 05292000 | Minnesota River at Ortonville, MN | 1939–2013 | 75 | Mainstem river |
| 05304500 | Chippewa River near Milan, MN | 1938-2013 | 76 | |
| 05311000 | Minnesota River at Montevideo, MN | 1930-2013 | 84 | Mainstem river |
| 05313500 | Yellow Medicine River near Granite Falls, MN | 1940-2013 | 74 | |
| 05315000 | Redwood River near Marshall, MN | 1941-2013 | 73 | |
| 05316500 | Redwood River near Redwood Falls, MN | 1936-2013 | 78 | |
| 05317000 | Cottonwood River near New Ulm, MN | 1939-2013 | 75 | |
| 05320000 | Blue Earth River near Rapidan, MN | 1950-2013 | 64 | |
| 05320500 | Le Sueur River near Rapidan, MN | 1950-2013 | 64 | |
| 05325000 | Minnesota River at Mankato, MN | 1930-2013 | 84 | Mainstem river |
| 05330000 | Minnesota River near Jordan, MN | 1935-2013 | 79 | Mainstem river - Downstream gauge |
| *Red River of the North Basin (22 gauges)* | | | | |
| 05050000 | Bois de Sioux River near White Rock, SD | 1942-2013 | 72 | |
| 05046000 | Otter Tail River near Fergus Falls, MN | 1931-2013 | 83 | |
| 05051500 | Red River of the North at Wahpeton, ND | 1944-2013 | 70 | Mainstem river |
| 05053000 | Wild Rice River near Abercrombie, ND | 1933-2013 | 81 | |
| 05056000 | Sheyenne River near Warwick, ND | 1950-2013 | 64 | |
| 05057000 | Sheyenne River near Cooperstown, ND | 1945-2013 | 69 | |
| 05058000 | Sheyenne River below Baldhill Dam, ND | 1950-2013 | 64 | |
| 05059000 | Sheyenne River near Kindred, ND | 1950-2013 | 64 | |
| 05059500 | Sheyenne River at West Fargo, ND | 1930-2013 | 84 | |
| 05054000 | Red River of the North at Fargo, ND | 1929-2013 | 85 | Mainstem river |
| 05060500 | Rush River at Amenia, ND | 1947-2013 | 67 | |



| 05062000 | Buffalo River near Dilworth, MN | 1932-2013 | 82 | |
| 05066500 | Goose River at Hillsboro, ND | 1935-2013 | 79 | |
| 05064000 | Wild Rice River at Hendrum, MN | 1945-2013 | 67 | Missing data: 1984 - 1985 |
| 05069000 | Sand Hill River at Climax, MN | 1947-2013 | 65 | Missing data: 1984 - 1985 |
| 05074500 | Red Lake River near Red Lake, MN | 1934-2013 | 74 | Missing data: 1994 - 1999 |
| 05075000 | Red Lake River at High Landing near Goodridge, MN | 1930-1999 | 70 | Missing data: 2000 - 2013 |
| 05076000 | Thief River near Thief River Falls, MN | 1929-2013 | 83 | Missing data: 1981 - 1982 |
| 05078000 | Clearwater River at Plummer, MN | 1940-2013 | 70 | Missing data: 1979 - 1982 |
| 05078500 | Clearwater River at Red Lake Falls, MN | 1935-2013 | 77 | Missing data: 1981 - 1982 |
| 05079000 | Red Lake River at Crookston, MN | 1929-2013 | 85 | |
| 05082500 | Red River of the North at Grand Forks, ND | 1929-2013 | 85 | Mainstem river - Downstream gauge |

† Tributary gauges, predominantly agricultural, not influenced by major dams
* Mean Annual Flow and Seven Day Low Flow Winter 1949-2013

**Table 2. Site details for AmeriFlux sites used for comparison with Livneh et al. (2013) evapotranspiration data. Average annual difference is positive when L13/L13(JJA) ET is greater than Ameriflux ET and negative when less than.**

| Site name | Willow Creek, WI | Bondville, IL | Rosemount, MN | Brookings, SD |
|---|---|---|---|---|
| **AmeriFlux site no.** | US-WCr | US-Bo1 | US-Ro1 | US-Bkg |
| **Latitude** | 45.8059 | 40.0062 | 44.7143 | 44.3453 |
| **Longitude** | -90.0799 | -88.2904 | -93.0898 | -96.8362 |
| **Nearest watershed[s]** | CRB | IRB | MRB [CRB] | MRB [RRB] |
| **Distance to nearest watershed (km)** | 0.463 | 13.049 | 43.807 [74.169] | 25.949 [129.688] |
| **Years** | 1999-2002 | 2003-2008 | 2004-2009 | 2004-2009 |
| **Vegetation** | Deciduous broadleaf forest | Croplands | Croplands | Grasslands |
| **Average difference L14-Ameriflux** | +31% | +17% | +14% | -29% |
| **Average difference L14(JJA)-Ameriflux** | +19% | +7% | +5% | -34% |

5    **Table 3. Summary of the breakpoint years identified from LCT (Fig. 3), piecewise linear regression (PwLR) of precipitation (P) and streamflow (Q), and continuous wavelet transform of P and Q (Fig. 4).**

| | LCT (Fig. 3) | P (PwLR) | Q (PwLR) | P ( CWT, Fig. 4) | Q (CWT, Fig. 4) |
|---|---|---|---|---|---|
| **RRB** | 2003/2004 | 1987/1988 | 1989/1990 | No change | 1995 |
| **MRB** | 1978/1979 | 1958/1959 | 1967/1968 | No change | 1980 |
| **IRB** | 1961/1962 | 1981/1982 | 1981/1982 | No change | 1975 |
| **CRB** | 2009/2010 | 1995/1996 | 1995/1996 | No change | No change |



**Table 4. Observed average annual precipitation (P), flow (Q), evapotranspiration (ET) and storage $\left(\frac{dS}{dt}\right)$ depths (cm y⁻¹) for each basin during the pre-period (a) and post-period (b) split by 1974/1975 (1) and LCT (2) breakpoints.**

| | | Years | $P_{mean}$ (cm y⁻¹) | $Q_{mean}$ (cm y⁻¹) | $ET_{mean}$ (cm y⁻¹) | $\frac{dS}{dt}_{mean}$ (cm y⁻¹) |
|---|---|---|---|---|---|---|
| MRB | 1a | 1935-1974 | 65.1 | 7.2 | 60.9 | -3.0 |
| | 1b | 1975-2011 | 70.0 | 13.4 | 64.2 | -7.5 |
| | 2a | 1935-1978 | 64.8 | 7.0 | 60.6 | -2.8 |
| | 2b | 1979-2011 | 71.0 | 14.4 | 65.0 | -8.4 |
| | 1a† | 1935-1974 | | | 55.6 | 2.3 |
| | 1b† | 1975-2011 | | | 58.7 | -2.0 |
| | 2a† | 1935-1978 | | | 55.4 | 2.4 |
| | 2b† | 1979-2011 | | | 59.3 | -2.7 |
| RRB | 1a | 1935-1974 | 53.4 | 3.7 | 45.1 | 4.7 |
| | 1b | 1975-2011 | 57.7 | 6.7 | 47.4 | 3.5 |
| | 2a | 1935-2003 | 54.5 | 4.6 | 45.6 | 4.4 |
| | 2b | 2004-2011 | 63.3 | 10.1 | 51.6 | 1.5 |
| | 1a† | 1935-1974 | | | 41.1 | 8.6 |
| | 1b† | 1975-2011 | | | 43.3 | 7.6 |
| | 2a† | 1935-2003 | | | 41.6 | 8.4 |
| | 2b† | 2004-2011 | | | 47.4 | 5.8 |
| IRB | 1a | 1939-1974 | 90.5 | 27.3 | 73.2 | -10.0 |
| | 1b | 1975-2011 | 95.2 | 33.0 | 75.1 | -13.0 |
| | 2a | 1939-1961 | 89.5 | 25.9 | 72.8 | -9.3 |
| | 2b | 1962-2011 | 94.4 | 32.2 | 74.8 | -12.5 |
| | 1a† | 1939-1974 | | | 66.9 | -3.7 |
| | 1b† | 1975-2011 | | | 68.7 | -6.6 |
| | 2a† | 1939-1961 | | | 66.5 | -3.0 |
| | 2b† | 1962-2011 | | | 68.4 | -6.1 |
| CRB | 1a | 1935-1974 | 80.0 | 29.7 | 61.8 | -11.5 |
| | 1b | 1975-2011 | 82.1 | 29.8 | 62.7 | -10.5 |
| | 2a | 1935-2009 | 80.8 | 29.6 | 62.1 | -11.0 |
| | 2b | 2010-2011 | 88.4 | 33.3 | 68.5 | -13.4 |
| | 1a† | 1935-1974 | | | 56.5 | -6.2 |
| | 1b† | 1975-2011 | | | 57.4 | -5.2 |
| | 2a† | 1935-2009 | | | 56.8 | -5.7 |
| | 2b† | 2010-2011 | | | 62.3 | -7.3 |

† 17% reduction in ET during summer months (JJA)