# Peer review of "Human amplified changes in precipitation-runoff patterns in large river basins of the Midwestern United States"

_Hydrology and Earth System Sciences, 2017_

## Referee Comment (RC1) · B. Livneh (Referee) · 2 Apr 2017

Overview

The authors address the interesting problem of disentangling anthropogenic versus climate impacts on hydrology in agricultural catchments in the mid-western US. They propose that storage has decreased dramatically in drained (tile) watersheds and discuss other aspects of the water budget, as well as conduct a break-point analysis to understand drivers of LCLUC changes. Overall, this is a wonderful analysis and the most interesting paper I've read in a while, so I'd like to commend the authors on a clearly articulated and thoughtful manuscript. A few points need to be clarified. However, I find the manuscript to be suitable for publication after minor revisions.

[Figure]

Major points

INTRO, P2 second paragraph: do the widely reported systematic increases in peak, mean, total, and base flows from the literature attribute these to decreases in ET, or solely from increases in precipitation? This point needs to be clarified and discussed further.

P3L20: studies report reductions in early season ET—presumably these are because replacing mature grasslands with fledgling crops reduces ET early in the season. However, what occurs later in the season, when the crops mature—will the ET be greater than grasslands?

What is the spatial resolution of the census drainage data? For which 5 years are drainage data available at the county-level?

The use of the Livneh et al. hydrometeorology data allows for calculation of the water balance at scales that are appropriate for the analysis. Although the authors acknowledge that the derived hydrologic outputs, e.g. ET, were generated using a modeling framework that considered static vegetation cover, they should report (if possible), which vegetation cover was used in VIC, e.g. was it natural vegetation or crop land cover? This would bolster the authors acknowledgement of the limitation.

Would the use of static land cover of Livneh et al. (2013) mean that the authors results are a conservative estimate of LCLUC impacts, or would this mean that the authors findings would overestimate impacts?

It would be useful to see a figure that shows historical land-cover change, precipitation change, and streamflow through time, if it is straightforward to show these together, as this would be very informative.

Would it be possible to test the interpretation hypothesis (2) in the discussion, that precipitation intensity may be influencing runoff efficiency? This could be something for future work, but would be an interesting experiment.

[Figure]

Minor points

I don't think "Midwestern" is a technical term, rather the Northeaster Great Plains is probably more apt and the authors should consider revising the references and title accordingly.

How did the authors reach the number of 286 for the t-test and KS-tests? This needs to be clarified as it is presently unclear.

All figures—it is unacceptable to include acronyms in the figure and then not define them in the caption. The figures should be readable as standalones. Hence, the authors need to define all acronyms in each figure in the respective captions.

Figure 5, explain briefly how the flow was normalized in the caption.

---

## Author Comment (AC1) · 28 Apr 2017

First we would like to thank referee Ben Livneh for reviewing our manuscript and providing constructive feedback. Original comments by the referee are denoted by "Referee Comment" and our responses are denoted by "Author Response".

Referee Comment: Overview The authors address the interesting problem of disentangling anthropogenic versus climate impacts on hydrology in agricultural catchments in the mid-western US. They propose that storage has decreased dramatically in drained (tile) watersheds and discuss other aspects of the water budget, as well as conduct a break-point analysis to understand drivers of LCLUC changes. Overall, this is a wonderful analysis and the most interesting paper I've read in a while, so I'd like to

commend the authors on a clearly articulated and thoughtful manuscript. A few points need to be clarified. However, I find the manuscript to be suitable for publication after minor revisions.

Author Response: Thank you! We are thrilled to hear that you find our analysis interesting and well-articulated.

Referee Comment: Major points INTRO, P2 second paragraph: do the widely reported systematic increases in peak, mean, total, and base flows from the literature attribute these to decreases in ET, or solely from increases in precipitation? This point needs to be clarified and discussed further.

Author Response: Increases in streamflows reported on page 2, lines 8-12, have been attributed to the combined effects of increasing precipitation and decreased ET from land use changes, including agricultural tile drainage and replacement of perennial vegetation and/or hay and small grains to corn and soybean rotations.

For example, Frans et al. 2013 examined the relative contributions of increasing precipitation and land use land cover change to observed streamflows in the Upper Mississippi River Basin (UMRB), upstream of Grafton, IL. They show that ET is expected to increase with twentieth century agricultural expansion, except in the places they modeled agricultural tile drainage. When tile drainage is present, ET decreases, while total runoff increases. This is entirely consistent with what we propose in our manuscript. Necessarily, storage must decrease between the pre and post period in the agricultural river basins to explain modern day water budgets and streamflow patterns. Tile drainage can accomplish this decrease in storage by draining soil moisture that would have otherwise gone as ET or contribute to regional groundwater. Therefore, twentieth century tile drainage expansion is expected to decrease ET and increase total runoff.

Schottler et al. 2014 corroborated this finding, and developed an empirical relationship between water yield and amount of precipitation (P) that goes as potential evapotranspiration (PET), PET/P. Their findings suggest that the PET/P ratio has decreased during the twentieth century due to combined effects of climate and crop conversions, and has contributed to the observed increases in annual water yields.

We discuss changes in annual runoff ratios, precipitation, and evapotranspiration in section 4.2.1 (specifically p. 17, lines 7-14), and will discuss ET findings of other streamflow change studies further in the introduction of the revised manuscript, as recommended by the reviewer.

Referee Comment: P3L20: studies report reductions in early season ETâĚŸAĚĞTpresumably these are because replacing mature grasslands with fledgling crops reduces ET early in the season. However, what occurs later in the season, when the crops matureâĚŸAĚĞTwill the ET be greater than grasslands?

Author Response: Although studies generally agree that conversion of mature prairie or grasslands with annual row crops reduce ET early in the growing season, there are mixed findings about how this land cover conversion affects ET later in the growing season, as well as annually (p. 2, lines 16-19). Crop growth and water use (ET) are highly dependent on local antecedent conditions such as precipitation, wind, humidity, solar radiation, and crop growth stage. For example, Zeri et al. (2013) found that maize had the highest values of ET annually in 2009 but the lowest values of ET in the drought year 2011, when compared to water use by miscanthus, switchgrass, and native prairie in central Illinois. In general, total annual water use between annual row crops and native prairie are not drastically different in Iowa (Wolf and Market 2007). However the distribution of water use throughout the season may be differ depending on antecedent climate conditions, as well as crop planting, emergence, and harvesting date. Because row crops have a relatively short growing season – planted generally in late April through early June, maximum growth and water use generally occurring in July-August , and harvested in September-October – evapotranspiration rates can be greater than native prairie during the peak growing season (July/August) and less than native prairie during early spring and late fall (Wolf and Market 2007). We will clarify this point in our revised manuscript.

Referee Comment: What is the spatial resolution of the census drainage data? For which 5 years are drainage data available at the county-level?

Author Response: The census drainage data are reported at the county level for 1940, 1950, 1960, 1978, and 2012 (page 6, lines 14-16). These are, unfortunately, the best available data for this spatial extent.

Referee Comment: The use of the Livneh et al. hydrometeorology data allows for calculation of the water balance at scales that are appropriate for the analysis. Although the authors acknowledge that the derived hydrologic outputs, e.g. ET, were generated using a modeling framework that considered static vegetation cover, they should report (if possible), which vegetation cover was used in VIC, e.g. was it natural vegetation or crop land cover? This would bolster the authors acknowledgement of the limitation.

Author Response: In a previously submitted version of the paper (doi:10.5194/hess-2016-571, p. 28, lines 7-13) we discussed the limitations associated with using the Hansen et al. (2000) static global vegetation classification in the VIC model. Several referees suggested significant shortening of the manuscript. Upon our own review, we eliminated details (∼2600 words) that were not essential to the manuscript. However, we agree that this would be a useful piece of information to convey for readers interested in this level of detail, so we will include this information in the Supplement of the revised manuscript.

Referee Comment: Would the use of static land cover of Livneh et al. (2013) mean that the authors results are a conservative estimate of LCLUC impacts, or would this mean that the authors findings would overestimate impacts?

Author Response: As stated above, we originally discussed potential limitations of using the Livneh et al. (2013) evapotranspiration data in a previously submitted version of the paper and will consider including such discussions in the Supplement of the revised manuscript, specifically in discussion of Figures S4 and S5. In general, static vegetation that does not include tile drainage should mean the Livneh et al. (2013)

ET estimates are overestimated in croplands, especially during modern times. This is exactly what we found when we compared the ET estimates to nearby Ameriflux stations in cropland cover (Figure S5). This potential bias is what allows us, independent of the climate drivers of ET change, to test whether drainage affects water balances. We anticipate that incorporating dynamic vegetation and tile drainage expansion in the VIC model would have reduced ET estimates and allowed for water budget closure in our analysis (i.e. storage term = zero). That said, Frans et al. (2013) tested the effects of dynamic vs. static cropland cover and found no statistically significant results of this effect on modeled annual runoff. Given that ET estimates between cropland and prairie are relatively similar, especially at annual scales, we do not think that dynamic land cover alone would have fully explained our water budget storage deficits, unless tile drainage was explicitly included.

Referee Comment: It would be useful to see a figure that shows historical land-cover change, precipitation change, and streamflow through time, if it is straightforward to show these together, as this would be very informative.

Author Response: While we appreciate this suggestion and have considered creating such a figure, the paper already contains ten figures, and we believe that incorporating the three suggested metrics into a single figure may become too cluttered for interpretation. We gladly welcome further suggestions from the referee as to how we might create such a figure, but our opinion is that the information is most effectively shown as three separate plots.

Referee Comment: Would it be possible to test the interpretation hypothesis (2) in the discussion, that precipitation intensity may be influencing runoff efficiency? This could be something for future work, but would be an interesting experiment.

Author Response: We agree that this would be a wonderful line of inquiry for future work, however this type of analysis should be written as a separate paper.

Referee Comment: Minor points I don't think "Midwestern" is a technical term, rather

the Northeaster Great Plains is probably more apt and the authors should consider revising the references and title accordingly.

Author Response: While Midwestern may not be a formal ecoregion or physiographic province, the term is commonly used in academic literature to describe the large part of the US that is covered in our analysis. We believe it more effectively conveys the location to our audience than would the term Northeaster Great Plains.

Referee Comment: How did the authors reach the number of 286 for the t-test and KS-tests? This needs to be clarified as it is presently unclear.

Author Response: Good catch. Thank you for the careful eye! We regret the error made on page 9, line 25, which should read "312 t-tests and 312 KS-tests...for a total of 652 statistical tests". On page 9, lines 16-17, we state that we ran all statistical tests using three defined breakpoints for each basin: three breakpoints X four study basins X 13 (or 12 monthly values + 1 annual value) = 156 t-tests and 156 KS-tests for each precipitation (P) and streamflow (Q) record, which is how we arrived at 312 t-tests and 312-KS-tests. Finally, 312+312+28 = 652 statistical tests total. This point will be clarified in the revised manuscript.

Referee Comment: All figuresâËŸAËĞ Tit is unacceptable to include acronyms in the figure and then not define them in the caption. The figures should be readable as standalones. Hence, the authors need to define all acronyms in each figure in the respective captions.

Author Response: We would like to thank the referee for the suggested comment and will define acronyms in individual figure captions.

Referee Comment: Figure 5, explain briefly how the flow was normalized in the caption.

Author Response: We would like to thank the referee for the suggested comment and will define "Normalized Flow" in the figure caption.

References

Frans, C., Istanbulluoglu, E., Mishra, V., Munoz-Arriola, F. and Lettenmaier, D. P.: Are climatic or land cover changes the dominant cause of runoff trends in the Upper Mississippi River Basin?, Geophys. Res. Lett., 40(6), 1104–1110, doi:10.1002/grl.50262, 2013.

Hansen, M. C., DeFries, R. S., Townshend, J. R. and Sohlberg, R.: Global land cover classification at 1 km spatial resolution using a classification tree approach, Int. J. Remote Sens., 21(6-7), 1331–1364 [online] Available from: http://data.globalforestwatch.org/datasets/7876b225f8034a0ebba79fad4afb80ad, 2000.

Livneh, B., Rosenberg, E. A., Lin, C., Nijssen, B., Mishra, V., Andreadis, K. M., Maurer, E. P. and Lettenmaier, D. P.: A Long-Term Hydrologically Based Dataset of Land Surface Fluxes and States for the Conterminous United States: Update and Extensions, J. Clim., 26, 9384–9392, 2013.

Schottler, S. P., Ulrich, J., Belmont, P., Moore, R., Lauer, J. W., Engstrom, D. R. and Almendinger, J. E.: Twentieth century agricultural drainage creates more erosive rivers, Hydrol. Process., 28(4), 1951–1961, doi:10.1002/hyp.9738, 2014.

Wolf, R.A., and Market, P.S.: On the impact of corn and soybeans to the local moisture budget in Iowa, National Weather Digest, 31(1), 3-7, 2007.

Zeri, M., Hussain, M. Z., Anderson-Teixeira, K. J., Delucia, E. and Bernacchi, C. J.: Water use efficiency of perennial and annual bioenergy crops in central Illinois, J. Geophys. Res. Biogeosciences, 118(2), 581–589, doi:10.1002/jgrg.20052, 2013.

———————————————

---

## Referee Comment (RC2) · Anonymous Referee #2 · 30 Apr 2017

This paper analyses observed changes in streamflow patters in four large basins of the Midwestern Unites States, and investigate their association with changes in climate (precipitation and evapotranspiration) and changes in land-use and land-cover, specifically the increasing cultivation of soybean and corn enhanced by artificial drainage. By analysing 79 years (1935 - 2013) of precipitation, streamflow, artificial drainage, and cultivation data, the authors provide a comprehensive statistical time series analysis to identify breakpoint years that could show relations between changes in magnitude and trends of the variables. Also, through the application of a simple water budget, the changes in basin storage are associated to changes in climate or in land-cover affecting streamflow response. The study concludes that artificial drainage as part of large

agricultural development in the Midwestern US amplifies the changes in rainfall-runoff response because of an increased hydrologic connectivity and a reduced storage in the basin.

The paper is interesting, and the analysis is extensive and well structured, so it should be suitable por publication after revisions. Although the authors have reported a reduction in word count with respect to a previous version, some parts of the document could be paraphrased and made more concise. Overall, the methods applied are robust and the results are well described and presented. However, there are two issues that, I think, need to be attended by the authors, which are the reasons for recommending revisions and re-review:

Major comments:

- The results section is an extensive description of the figures and tables, but less of an actual discussion and analysis. For example, the authors present all the numbers for each individual catchment repeatedly, but they don't compare these results with similar studies or discuss them in a broader context for interpretation. As this section is the main and longest part of the paper, after such a comprehensive exposition of results, the reader may feel disappointed not to find an equivalent discussion and interpretation of results. I would suggest to divide the section between Results (from a purely descriptive perspective only) and Discussion and Interpretation (extending on the current last section) to highlight better the value of this study and the findings. The closing Conclusions section could be short and concise as well.

- The study uses 7 metrics to analyse the streamflow regime, but it is not clear how and why these indices were selected. The literature is quite extensive with respect to hydrological indices to characterise and analyse streamflow features and alterations. See for example: Olden and Poff (2003); and Ochoa-Tocachi et al. (2016) for a list of indices used extensively in hydrological studies. The metrics (indices) depend on the streamflow attributes that the study is investigating, and ideally able to represent

different parts of the hydrological response (independent, non-redundant) to provide more holistic views. The paper mostly focuses on streamflow magnitude, but there are other attributes (frequency, duration, timing, and flashiness) of flows that could be of interest. Lastly, it is not clear how the several streamflow gauges were used or, at least, the value of using several gauges in contrast to the downstream outlet only is lost.

Specific comments:

P1L27: See other attributes of streamflow: magnitude, frequency, duration, timing, flashiness.

P2L8-12: This is an example of a long sentence that could be divided or reduced.

P2L15: Specify if the term runoff refers to overland flow or total streamflow.

P2L21: Try not to cite articles not published yet.

P3L6: Although the term "artificial drainage" is used several times in the first subsection, it is only defined at this point. Maybe you could move the definition to the first time the term is mentioned.

P3L18: As part of another long sentence, it is unclear if the phrase between the commas ", at least for well drained soils (Hamilton et al., 2015)," refers to a study that does not show a reduction in ET, or if this is a condition for the following studies that report such reduction.

P5L3: The acronym PRISM is used here, but only introduced in the next section. Try to define acronyms the first time they are mentioned.

P5L21: When referring to tile installation, clarify that it is the "ANNUAL installation (or installation RATE) [which] has increased from 3 miles in 1999 to 1,924 miles in 2015".

P6L4: Generally the acronym for land-use and land-cover changes is (LUCC). However, as this term is widely used across the paper, check what is the most common term in the literature for your potential readers, if you want to keep LULC change as it

is.

P9L11: As the authors mention, when fo»0, the Morlet wavelet is simplified. Is the value of fo=0.849 considered much greater than 0?

P18L3-8 and P18L27-P19L2: These two groups of sentences should not be part of the results but of the methods as they explain how the figures must be read. This is an example of how the results section could be shortened.

P21L3: Flashiness (and flashy) is actually an attribute of streamflow, so no quotation marks are needed. The term "rate of change" of flows can also be used. Check the comment on the selection of metrics.

References:

Olden, J. D., and N. L. Poff (2003), Redundancy and the choice of hydrologic indices for characterizing streamflow regimes, River Res. Appl., 19(2), 101 - 121, doi:10.1002/rra.700.

Ochoa-Tocachi, B. F., W. Buytaert, and B. De Bievre (2016), Regionalization of land-use impacts on streamflow using a network of paired catchments, Water Resour. Res., 52, 6710 - 6729, doi:10.1002/2016WR018596.v
* * *

---

## Referee Comment (RC3) · Anonymous Referee #3 · 15 May 2017

The authors use a variety of methods to characterize the changes in hydrology in four large Upper Midwest USA watersheds due to precipitation and land cover change. The focus of the work is relevant to the region and deserving of publication. The paper is well-written but several points are worthy of additional attention: 1. Page 2, Lines 15-30- The authors insist that the main issue associated with increased flows and base-flows is increasing sediment loads. This is an issue very close to their working group in Minnesota. However, for the rest of the Midwest, the issue is increasing nutrient loads, specifically nitrate and phosphorus. Tile drainage is the main source of nitrate to rivers and it is barely mentioned in the paper. Outside of the Minnesota River, tile drainage is rarely mentioned along with bank erosion but the issue of tile drainage is universally

considered a dominating factor in nitrate and dissolved phosphorus transport. The authors should change their focus to include more discussion of the relevance of increasing flows on nutrient export and Gulf Hypoxia. There is a wealth of papers on this topic that can be considered and they are largely ignored in the paper. 2.Likewise the authors ignore research conducted on this issue previously including: Xu, Xianli, et al. "Relative importance of climate and land surface changes on hydrologic changes in the US Midwest since the 1930s: Implications for biofuel production." Journal of Hydrology 497 (2013): 110-120. This paper used some different methods to derive some assessment of the topic. There are other papers where this came from. The topic is not new and the authors should compare their results to the body of literature reporting on the same topic. 3. Page 5, lines 7-23 - The authors again ignore a body of research on the extent of tile drainage in the US Midwest (search for papers from Mark David in JEQ). There are much better estimates of tile drainage extent available than NASS. What is the source of the percentages in line 19? 4. Page 5 line 3 - PRISM used without definition; Page 7 lines 13 and 27 it is defined twice. 5. Page 8 CWT methods - there is a lot of method text devoted to this method but it did not prove add much to the results and discussion. In fact it is largely ignored later in the paper in a single short paragraph. I would suggest dropping this method or simply mentioning that it was done and moving to supplemental text. It adds nothing to your argument. 6. Page 9, line 12-19 - the rationale for the breakpoint based on plastic pipe is purely speculative and it gives this idea credence. Just simply break the record up in two equal periods consistent with previous work. 7. Where did the ET data for crops come from? The water balance method is hugely sensitive to ET and it seems rather arbitrary to reduce it by 17% because of a literature citation. This needs to be verified independently by the authors. Maybe its 25% or 10%, who knows, and yet it is retained in the results and discussion like there is true meaning behind it. 8. Section 4.1 - this seems like background material to me. You didn't really do anything new here except compile some data available in databases. Again, the NASS data is pretty weak for these trends, especially trends reported as if they are completely accurate. This is qualitative data at best, 9.Page 6,

lines 16-21 - it is a stretch to cite Figure 5 to discuss cyclicity. It is not obvious in the figure. suggest that the authors find a way to make it visible or drop from the text. 10. I found the data is Figure 6 to be the most compelling in the paper. Are the deviations consistent with the imposed breakpoints? 11. Section 4.2.3 - the use of daily scale in this multi-year analysis is inappropriate. The results are very weak and do not really add to your argument. What about the "flashiness index"? There is no real link of this section to your main issues and I would suggest dropping this section.

Overall, the paper adds to the body of research that already exists on the topic. The topic is not new and it has been evaluated by many authors previously but there are some methods and techniques used and the issue of worthy of attention. I am recommending the paper be accepted with major revisions.

---

## Referee Comment (RC4) · Anonymous Referee #4 · 19 May 2017

This paper treats an interesting topic: the effect of human activities on the precipitation-runoff patters in the Midwestern United States. In the last century, the land transformation from natural to agriculture and urban areas seriously affected changes in the hydrology of the study area. The results suggested that storage has decreased in intensively drained and cultivated basins by 30%-200% since 1975, but increased by 30% in the less agricultural basin. This has amplified the streamflow response to precipitation increases in the Midwest.

While the results are quite interesting some important information is obscured or not well described:

- infiltration and hydraulic soil properties (e.g. spatial data);

- water storage capacity of ditches (what kind of ditches are these? Only surface drainage system? What about the sub-surface drainage network?);

- methodology to recognize/map ditches (the authors highlighted some issues in the underestimation of their extent; is this issue critical for the suitability of the final results?);

- surface runoff related to the different agriculture practices through years.

Also, some recent relevant papers related to the drainage ditch role in water storage capacity, effects of the changes in drainage ditches density and agricultural practices on water storage capacity are missed in the literature review.

Unfortunately, because of the above critical issues, the paper needs to be restructured adding extra info to provide a more clear view of the processes involved in the study area.

---

## Author Response (AR1)

**Page and line numbers in point-by-point replies refer to the version of the paper published on 15 Mar 2017 unless otherwise stated, such as those in the margin comments. 15 Mar 2017 paper version available at: https://doi.org/10.5194/hess-2017-133**

5   **Section 1.a - Author Response to Referee #1: Ben Livneh (previous version published in HESS online discussion 28 April 2017: doi:10.5194/hess-2017-133-AC1, 2017)**

First we would like to thank referee Ben Livneh for reviewing our manuscript and providing constructive feedback. Original comments by the referee are denoted by "Referee Comment" and our responses are denoted by "Author Response".

10   Referee Comment: Overview The authors address the interesting problem of disentangling anthropogenic versus climate impacts on hydrology in agricultural catchments in the mid-western US. They propose that storage has decreased dramatically in drained (tile) watersheds and discuss other aspects of the water budget, as well as conduct a break-point analysis to understand drivers of LCLUC changes. Overall, this is a wonderful analysis and the most interesting paper I've read in a while, so I'd like to commend the authors on a clearly articulated and thoughtful manuscript. A few points need to

15   be clarified. However, I find the manuscript to be suitable for publication after minor revisions.

Author Response: Thank you! We are thrilled to hear that you find our analysis interesting and well-articulated.

Referee Comment: Major points INTRO, P2 second paragraph: do the widely reported systematic increases in peak, mean,

20   total, and base flows from the literature attribute these to decreases in ET, or solely from increases in precipitation? This point needs to be clarified and discussed further.

> **Comment [SK1]:** See changes made in this document on page 20

Author Response: Increases in streamflows reported on page 2, lines 8-12, have been attributed to the combined effects of increasing precipitation and decreased ET from land use changes, including agricultural tile drainage and replacement of

25   perennial vegetation and/or hay and small grains to corn and soybean rotations.

For example, Frans et al. 2013 examined the relative contributions of increasing precipitation and land use land cover change to observed streamflows in the Upper Mississippi River Basin (UMRB), upstream of Grafton, IL. They show that ET is expected to increase with twentieth century agricultural expansion, except in the places they modeled agricultural tile drainage. When tile drainage is present, ET decreases, while total runoff increases. This is entirely consistent with what

30   we propose in our manuscript. Necessarily, storage must decrease between the pre and post period in the agricultural river basins to explain modern day water budgets and streamflow patterns. Tile drainage can accomplish this decrease in storage by draining soil moisture that would have otherwise gone as ET or contribute to regional groundwater. Therefore, twentieth century tile drainage expansion is expected to decrease ET and increase total runoff.

Schottler et al. 2014 corroborated this finding, and developed an empirical relationship between water yield and amount of precipitation (P) that goes as potential evapotranspiration (PET), PET/P. Their findings suggest that the PET/P ratio has decreased during the twentieth century due to combined effects of climate and crop conversions, and has contributed to the observed increases in annual water yields.

5    We discuss changes in annual runoff ratios, precipitation, and evapotranspiration in section 4.2.1 (specifically p. 17, lines 7-14), and have included ET findings of other streamflow change studies further in the introduction of the revised manuscript, as recommended by the reviewer.

Referee Comment: P3L20: studies report reductions in early season ETâËŸAËGTpre-˘sumably these are because replacing
10   mature grasslands with fledgling crops reduces ET early in the season. However, what occurs later in the season, when the crops matureâËŸAËGTwill the ET be greater than grasslands?˘

**Comment [SK2]:** See changes made in this document on page 20

Author Response: Although studies generally agree that conversion of mature prairie or grasslands with annual row crops reduce ET early in the growing season, there are mixed findings about how this land cover conversion affects ET later in the
15   growing season, as well as annually (p. 2, lines 16-19). Crop growth and water use (ET) are highly dependent on local antecedent conditions such as precipitation, wind, humidity, solar radiation, and crop growth stage. For example, Zeri et al. (2013) found that maize had the highest values of ET annually in 2009 but the lowest values of ET in the drought year 2011, when compared to water use by miscanthus, switchgrass, and native prairie in central Illinois. In general, total annual water use between annual row crops and native prairie are not drastically different in Iowa (Wolf and Market 2007). However the
20   distribution of water use throughout the season may be differ depending on antecedent climate conditions, as well as crop planting, emergence, and harvesting date. Because row crops have a relatively short growing season – planted generally in late April through early June, maximum growth and water use generally occurring in July-August , and harvested in September-October – evapotranspiration rates can be greater than native prairie during the peak growing season (July/August) and less than native prairie during early spring and late fall (Wolf and Market 2007). We have clarified this
25   point in our revised manuscript.

Referee Comment: What is the spatial resolution of the census drainage data? For which 5 years are drainage data available at the county-level?

**Comment [SK3]:** No change to manuscript.

30   Author Response: The census drainage data are reported at the county level for 1940, 1950, 1960, 1978, and 2012 (page 6, lines 14-16). These are, unfortunately, the best available data for this spatial extent.

Referee Comment: The use of the Livneh et al. hydrometeorology data allows for calculation of the water balance at scales that are appropriate for the analysis. Although the authors acknowledge that the derived hydrologic outputs, e.g. ET, were generated using a modeling framework that considered static vegetation cover, they should report (if possible), which vegetation cover was used in VIC, e.g. was it natural vegetation or crop land cover? This would bolster the authors acknowledgement of the limitation.

**Comment [SK4]:** See changes made in this document on page 26 and pages 62 & 63

Author Response: In a previously submitted version of the paper (doi:10.5194/hess- 2016-571, p. 28, lines 7-13) we discussed the limitations associated with using the Hansen et al. (2000) static global vegetation classification in the VIC model. Several referees suggested significant shortening of the manuscript. Upon our own review, we eliminated details (∼2600 words) that were not essential to the manuscript. However, we agree that this would be a useful piece of information to convey for readers interested in this level of detail, so we will include this information in the Supplement of the revised manuscript.

Referee Comment: Would the use of static land cover of Livneh et al. (2013) mean that the authors results are a conservative estimate of LCLUC impacts, or would this mean that the authors findings would overestimate impacts?

**Comment [SK5]:** See changes made in this document on page 26 and pages 62 & 63

Author Response: As stated above, we originally discussed potential limitations of using the Livneh et al. (2013) evapotranspiration data in a previously submitted version of the paper and have included such discussions in the Supplement of the revised manuscript, specifically in discussion of Figures S4 and S5. In general, static vegetation that does not include tile drainage should mean the Livneh et al. (2013) ET estimates are overestimated in croplands, especially during modern times. This is exactly what we found when we compared the ET estimates to nearby Ameriflux stations in cropland cover (Figure S5). This potential bias is what allows us, independent of the climate drivers of ET change, to test whether drainage affects water balances. We anticipate that incorporating dynamic vegetation and tile drainage expansion in the VIC model would have reduced ET estimates and allowed for water budget closure in our analysis (i.e. storage term = zero). That said, Frans et al. (2013) tested the effects of dynamic vs. static cropland cover and found no statistically significant results of this effect on modeled annual runoff. Given that ET estimates between cropland and prairie are relatively similar, especially at annual scales, we do not think that dynamic land cover alone would have fully explained our water budget storage deficits, unless tile drainage was explicitly included.

Referee Comment: It would be useful to see a figure that shows historical land-cover change, precipitation change, and streamflow through time, if it is straightforward to show these together, as this would be very informative.

**Comment [SK6]:** No change to manuscript.

Author Response: While we appreciate this suggestion and have considered creating such a figure, the paper already contains ten figures, and we believe that incorporating the three suggested metrics into a single figure may become too cluttered for interpretation. We gladly welcome further suggestions from the referee as to how we might create such a figure, but our opinion is that the information is most effectively shown as three separate plots.

Referee Comment: Would it be possible to test the interpretation hypothesis (2) in the discussion, that precipitation intensity may be influencing runoff efficiency? This could be something for future work, but would be an interesting experiment.

**Comment [SK7]:** No change to manuscript.

Author Response: We agree that this would be a wonderful line of inquiry for future work, however this type of analysis
10  should be written as a separate paper.

Referee Comment: Minor points I don't think "Midwestern" is a technical term, rather the Northeaster Great Plains is probably more apt and the authors should consider revising the references and title accordingly.

**Comment [SK8]:** No change to manuscript.

15  Author Response: While Midwestern may not be a formal ecoregion or physiographic province, the term is commonly used in academic literature to describe the large part of the US that is covered in our analysis. We believe it more effectively conveys the location to our audience than would the term Northeaster Great Plains.

Referee Comment: How did the authors reach the number of 286 for the t-test and KS-tests? This needs to be clarified as it is
20  presently unclear.

**Comment [SK9]:** See changes made in this document on page 28

Author Response: Good catch. Thank you for the careful eye! We regret the error made on page 9, line 25, which should read "312 t-tests and 312 KS-tests...for a total of 652 statistical tests". On page 9, lines 16-17, we state that we ran all statistical tests using three defined breakpoints for each basin: three breakpoints X four study basins X 13 (or 12 monthly
25  values + 1 annual value) = 156 t-tests and 156 KS-tests for each precipitation (P) and streamflow (Q) record, which is how we arrived at 312 t-tests and 312-KS-tests. Finally, 312+312+28 = 652 statistical tests total. This point will be clarified in the revised manuscript.

Referee Comment: All figuresâËŸAËG Tit is unacceptable to include acronyms in the ˘ figure and then not define them in
30  the caption. The figures should be readable as standalones. Hence, the authors need to define all acronyms in each figure in the respective captions.

**Comment [SK10]:** See tables, figures, and caption text changes in this document, below

Author Response: We would like to thank the referee for the suggested comment and will define acronyms in individual figure captions.

Referee Comment: Figure 5, explain briefly how the flow was normalized in the caption.

Author Response: We would like to thank the referee for the suggested comment and will define "Normalized Flow" in the figure caption.

**Section 1.b - Author Response to Anonymous Referee #2**

First we would like to thank Anonymous Referee #2 for reviewing our manuscript. Original comments by the referee are denoted by "Referee Comment" and our responses are denoted by "Author Response".

30  Referee Comment:  This paper analyses observed changes in streamflow patters in four large basins of the Midwestern Unites States, and investigate their association with changes in climate (precipitation and evapotranspiration) and changes in land-use and land-cover, specifically the increasing cultivation of soybean and corn enhanced by artificial drainage. By

analysing 79 years (1935 - 2013) of precipitation, streamflow, artificial drainage, and cultivation data, the authors provide a comprehensive statistical time series analysis to identify breakpoint years that could show relations between changes in magnitude and trends of the variables. Also, through the application of a simple water budget, the changes in basin storage are associated to changes in climate or in land-cover affecting streamflow response. The study concludes that artificial drainage as part of large agricultural development in the Midwestern US amplifies the changes in rainfall-runoff response because of an increased hydrologic connectivity and a reduced storage in the basin.

The paper is interesting, and the analysis is extensive and well structured, so it should be suitable por publication after revisions. Although the authors have reported a reduction in word count with respect to a previous version, some parts of the document could be paraphrased and made more concise. Overall, the methods applied are robust and the results are well described and presented. However, there are two issues that, I think, need to be attended by the authors, which are the reasons for recommending revisions and re-review:

Author Response: Thank you for finding our paper interesting and suitable for publication after revision. The suggested revisions are very much appreciated.

Major comments:

Referee Comment: The results section is an extensive description of the figures and tables, but less of an actual discussion and analysis. For example, the authors present all the numbers for each individual catchment repeatedly, but they don't compare these results with similar studies or discuss them in a broader context for interpretation. As this section is the main and longest part of the paper, after such a comprehensive exposition of results, the reader may feel disappointed not to find an equivalent discussion and interpretation of results. I would suggest to divide the section between Results (from a purely descriptive perspective only) and Discussion and Interpretation (extending on the current last section) to highlight better the value of this study and the findings. The closing Conclusions section could be short and concise as well.

**Comment [SK12]:** See changes made to manuscript in sections 4.1, 4.2.1, and 5 of the revised manuscript

Author Response: We appreciate the stylistic suggestion to separate the results and discussion sections. We have considered making this change, however, we present six distinct sets of analyses, each addressing specific questions. Therefore we believe the text is easier for the reader to follow if the results and discussion are presented together within each section. We do compare individual results to findings from other studies, e.g. p. 15, lines 14-16; p. 16, lines 28-31; p. 17, lines 10-12. We have removed a few sentences from the conclusion section, but in general believe that it provides a concise synthesis of our findings, a concise explanation of alternate hypotheses that can be pursued by future work, and a brief statement regarding the implications for policy and management.

Referee Comment: The study uses 7 metrics to analyse the streamflow regime, but it is not clear how and why these indices were selected. The literature is quite extensive with respect to hydrological indices to characterise and analyse streamflow features and alterations. See for example: Olden and Poff (2003); and Ochoa-Tocachi et al. (2016) for a list of indices used extensively in hydrological studies. The metrics (indices) depend on the streamflow attributes that the study is investigating, and ideally able to represent different parts of the hydrological response (independent, non-redundant) to provide more holistic views. The paper mostly focuses on streamflow magnitude, but there are other attributes (frequency, duration, timing, and flashiness) of flows that could be of interest. Lastly, it is not clear how the several streamflow gauges were used or, at least, the value of using several gauges in contrast to the downstream outlet only is lost.

> **Comment [SK13]:** No changes made to manuscript

Author Response: We appreciate your suggestion and we are aware that many metrics can be used to quantify hydrologic change. The seven streamflow metrics presented in Figure 5 are consistent with those used in previous studies of Midwestern agricultural basins, and capture properties of streamflow typically used for management and decision making (e.g., Novotny and Stefan, 2007; Vandegrift and Stefan, 2010). However, we go well beyond these seven streamflow metrics and present numerous analyses of precipitation and streamflow change at the monthly timescale (Figures 6-8) as well as daily timescale (Figure 9). Other magnitude-frequency metrics have been used in our previous study (Foufoula-Georgiou et al., 2015), such as probability density function (pdf) of the slopes of the rising and falling limbs of the hydrographs, metrics of non-linearity, multi-scale energy decomposition via wavelets, and joint pdf of precipitation and next-day streamflow, etc., but they are beyond the scope of the present study. Data presented in Figure 5 include data from numerous gages within each of the basins, as specified in the caption. However, all other analyses were conducted for the gages at the mouths of the major watersheds, as stated in section 3.3.

Specific comments:

Referee Comment: P1L27: See other attributes of streamflow: magnitude, frequency, duration, timing, flashiness.

> **Comment [SK14]:** See changes made to manuscript in this document, p. 20

Author Response: We have revised the statement to read "The magnitude, frequency, duration and timing of streamflows strongly influence water quality, sediment and nutrient transport, channel morphology, and habitat conditions of a river channel."

Referee Comment: P2L8-12: This is an example of a long sentence that could be divided or reduced.

> **Comment [SK15]:** See changes made to manuscript in this document, p. 20

Author Response: Thank you for the suggestion. We split the sentence in two.

Referee Comment: P2L15: Specify if the term runoff refers to overland flow or total streamflow.

Author Response: P2L15: We use the word runoff on page 2, line 15 to make a general statement acknowledging recent hydrologic changes in the Midwest. If we were referring to overland flow, we would have used that term. We believe that the
5 use of the word runoff is appropriate as written.

Referee Comment: P2L21: Try not to cite articles not published yet.

Author Response: Thank you for this comment. We included this as a placeholder, anticipating that the Lauer et al., paper
10 will be published before the HESS paper, which we still expect to be the case.

Referee Comment: P3L6: Although the term "artificial drainage" is used several times in the first subsection, it is only defined at this point. Maybe you could move the definition to the first time the term is mentioned.

15 Author Response: Thanks for the suggestion. We considered moving the definition earlier in the paper. However, the statements made in the previous section are all very high level points and inserting a specific definition negatively impacts the flow. Section 1.2 is focused on explaining the details of artificial drainage, so we believe it is the best place to provide this explanation.

20 Referee Comment: P3L18: As part of another long sentence, it is unclear if the phrase between the commas ", at least for well drained soils (Hamilton et al., 2015)," refers to a study that does not show a reduction in ET, or if this is a condition for the following studies that report such reduction.

Author Response: The phrase between the commas is meant to refer to the former statement. We have clarified this in the
25 revised manuscript by splitting the sentence in two.

Referee Comment: P5L3: The acronym PRISM is used here, but only introduced in the next section. Try to define acronyms the first time they are mentioned.

30 Author Response: Thank you for pointing this out. We deleted the term PRISM in this case as the important point is that we are talking about annual long term means. We cite the source of the data (and explain the acronym PRISM) in the figure caption as cited. PRISM is also explained now in its first usage in the text, at the beginning of section 3.2.

Referee Comment: P5L21: When referring to tile installation, clarify that it is the "ANNUAL installation (or installation RATE) [which] has increased from 3 miles in 1999 to 1,924 miles in 2015".

Comment [SK21]: See changes made to manuscript in this document, p. 24

5 Author Response: Thank you for this suggestion. We clarified that it is the annual installation that has increased. Additionally, we have converted the lengths to SI units. We initially used imperial units because those were the measurement units reported in the cited report. However, SI units are more appropriate for our global audience.

10 Referee Comment: P6L4: Generally the acronym for land-use and land-cover changes is (LUCC). However, as this term is widely used across the paper, check what is the most common term in the literature for your potential readers, if you want to keep LULC change as it is.

Comment [SK22]: No change made to manuscript

Author Response: Thank you for the suggestion. However, we would prefer to keep the acronym LULC, also used in our
15 previous work and many other papers in the literature. Also, we prefer not to incorporate "change" in the acronym. For example, we talk about LULC of a basin during different periods of time, and then infer "change" by comparison.

Referee Comment: P9L11: As the authors mention, when fo»0, the Morlet wavelet is simplified. Is the value of fo=0.849 considered much greater than 0?

Comment [SK23]: No change made to manuscript

Author Response: Yes, for any value of f0>0.8 the second term inside the parenthesis of equation (2), which corrects for the non-zero mean of the wavelet, becomes negligible, making (3) a proper wavelet of zero mean. The selection of the specific value of f0=0.849 (=1/sqrt(1/(2ln2))) is commonly used in practice as it produces a decay where the magnitude of the two peaks in the real waveform adjacent to the central peak are half its amplitude (see Addison, 2017, and also Appendix A in
25 Foufoula-Georgiou et al., 2015).

Referee Comment: P18L3-8 and P18L27-P19L2: These two groups of sentences should not be part of the results but of the methods as they explain how the figures must be read. This is an example of how the results section could be shortened.

Comment [SK24]: No change made to manuscript

30 Author Response: We respectfully disagree with the referee's suggestion that these statements should be move to the methods. Indeed, these statements help the reader understand what is being presented in the figures and how to read them. Therefore, we feel strongly that they are most appropriate and useful to the reader in the results section. We are not presenting new methods here, just reminding the reader of what these data represent.

Referee Comment: P21L3: Flashiness (and flashy) is actually an attribute of streamflow, so no quotation marks are needed. The term "rate of change" of flows can also be used. Check the comment on the selection of metrics.

**Comment [SK25]:** See changes made to manuscript in this document, p. 41

Author Response: Thank you for your suggestion. We have removed the quotation marks around the word flashiness here and elsewhere throughout the paper.

**Section 1.c - Author Response to Anonymous Referee #3**

First we would like to thank Anonymous Referee #3 for reviewing our manuscript and providing constructive feedback. Original comments by the referee are denoted by "Referee Comment" and our responses are denoted by "Author Response".

Referee Comment: The authors use a variety of methods to characterize the changes in hydrology in four large Upper Midwest USA watersheds due to precipitation and land cover change. The focus of the work is relevant to the region and deserving of publication. The paper is well-written but several points are worthy of additional attention:

Author Response: Thank you for your review of the paper, and for considering our work well written and relevant.

Referee Comment: 1. Page 2, Lines 15-30- The authors insist that the main issue associated with increased flows and base-flows is increasing sediment loads. This is an issue very close to their working group in Minnesota. However, for the rest of the Midwest, the issue is increasing nutrient loads, specifically nitrate and phosphorus. Tile drainage is the main source of nitrate to rivers and it is barely mentioned in the paper. Outside of the Minnesota River, tile drainage is rarely mentioned along with bank erosion but the issue of tile drainage is universally considered a dominating factor in nitrate and dissolved phosphorus transport. The authors should change their focus to include more discussion of the relevance of increasing flows on nutrient export and Gulf Hypoxia. There is a wealth of papers on this topic that can be considered and they are largely ignored in the paper.

Author Response: Thank you for pointing out this oversight. We agree that additional discussion of linkages to other water quality problems will greatly enhance the impact of the paper. We are aware of the vast literature on the topic (e.g. David et al., 1997; Goolsby et al., 1999; Kreiling and Houser, 2016; Letey et al., 1977; Randall and Mulla, 2001; Royer et al., 2006; Schilling et al., 2017; Sims et al., 1998 to name a few). We focused on sediment in part because it is the primary focus of our research group, as the reviewer points out, but also because there is so much known/written about the linkages between artificial drainage and nutrients and we believe it is important to highlight other indirect effects of tile drainage, such as increased bank erosion and thus fine sediment loads, which also degrade water quality in the US and globally. Understanding the many effects of tile drainage in agricultural landscapes underscores the continued need for better drainage management practices. Nutrients were already briefly mentioned in the abstract, introduction and conclusion. We have better clarified our attention to sediment and incorporated additional discussion of broader water quality concerns in the introduction and conclusion of the revised manuscript.

Referee Comment: 2.Likewise the authors ignore research conducted on this issue previously including: Xu, Xianli, et al. "Relative importance of climate and land surface changes on hydrologic changes in the US Midwest since the 1930s: Implications for biofuel production." Journal of Hydrology 497 (2013): 110-120. This paper used some different methods to derive some assessment of the topic. There are other papers where this came from. The topic is not new and the authors should compare their results to the body of literature reporting on the same topic.

Author Response: We are familiar with the work of Xu et al. 2013 and cited their work in previous versions of the paper. While revising and shortening the paper, we must have deleted statements citing their work. We apologize for this oversight.

We agree that the work by Xu et al. 2013 and others have contributed to the vast body of literature on the topic. We discuss how our findings compare to work by other authors on the topic – p. 15, lines 14-16; p. 16, lines 28-31; p. 17, lines 10-12. We have re-added the citation to Xu et al., 2013 and have bolstered our discussion of how our work compares to Xu et al. 2013 and other studies in the revised manuscript.

Referee Comment: 3. Page 5, lines 7-23 - The authors again ignore a body of research on the extent of tile drainage in the US Midwest (search for papers from Mark David in JEQ). There are much better estimates of tile drainage extent available than NASS. What is the source of the percentages in line 19?

**Comment [SK28]:** See changes made to manuscript in this document, p. 26

10 Author Response: We respectfully disagree with the referee's claim that "there are much better estimates of tile drainage extent available than NASS". In some states, estimates of tile drainage extent, taken from different sources, vary by a factor of two (Sugg, 2007). David et al. 2010, acknowledge that estimates of tile drainage extent can be made using a variety of GIS and aerial imagery approaches; however these methods are only feasible over small spatial extents, and usually have limited temporal resolution. Therefore, David et al. 2010 rely on county level, best guess estimates from Sugg (2007). We

15 discuss the limitations of the Sugg (2007) approach to making tile drainage estimates in the revised manuscript. However, we stand by our decision to report land use changes in each basin by reporting county-level drainage data from five US Census of Agriculture reports, as well as annual NASS crop cover data.

The source for the percentages in line 19 come from the USDA National Agricultural Statistics Service Cropland Data Layer. {2013}. Published crop-specific data layer [Online]. Available at https://nassgeodata.gmu.edu/CropScape/

20 (accessed {1 Oct 2016}; verified {19 May 2017}). USDA-NASS, Washington, DC. We have clarified and cited this source in the Figure 1 caption of our revised manuscript.

Referee Comment: 4. Page 5 line 3 - PRISM used without definition; Page 7 lines 13 and 27 it is defined twice.

**Comment [SK29]:** See page 26, line 12 in this document

25 Author Response: Thank you for your attention to detail! We have deleted PRISM from p. 5 ,line 3 as it is unnecessary to introduce here. Therefore, in the revised manuscript (see below in this document) PRISM is defined the first time it is mentioned, on p. 26, line 12.

Referee Comment: 5. Page 8 CWT methods - there is a lot of method text devoted to this method but it did not prove add

30 much to the results and discussion. In fact it is largely ignored later in the paper in a single short paragraph. I would suggest dropping this method or simply mentioning that it was done and moving to supplemental text. It adds nothing to your argument.

**Comment [SK30]:** See changes made to manuscript in this document, p. 28

Author Response: The referee is correct that we devote 18 lines to the CWT methods in section 3.4. We have shortened the text in this section of the revised manuscript, and removed one equation from the text.

5 Referee Comment: 6. Page 9, line 12-19 - the rationale for the breakpoint based on plastic pipe is purely speculative and it gives this idea credence. Just simply break the record up in two equal periods consistent with previous work.

Author Response: : We have used the term 'roughly' to indicate that while the date of widespread acceptance of corrugated plastic tile is not exactly know, the timing of our breakpoint is consistent with what is generally thought to be the transition, 10 as documented by Fouss and Reeve, 1987.

Referee Comment: 7. Where did the ET data for crops come from? The water balance method is hugely sensitive to ET and it seems rather arbitrary to reduce it by 17% because of a literature citation. This needs to be verified independently by the authors. Maybe its 25% or 10%, who knows, and yet it is retained in the results and discussion like there is true meaning 15 behind it.

Author Response: We used ET data from Livneh et al. 2013 (L13) for water balance calculations (p. 7, lines 18-24). We have, in fact, compared independent ET estimates from L13 to other sources in the Supplement (Figs. S4 & S5) and describe our rationale for applying the 17% reduction in JJA (page 10, lines 24-39). This adjustment does not change the 20 interpretations of the water budget; it simply represents our effort to err on the conservative side with the water budget (p. 23, lines 5-7). Livneh also reviewed our manuscript and did not object to this correction we applied. In any case, eliminating the 17% reduction would only strengthen the conclusion we reach, but we feel it is more reasonable to err on the conservative side with this calculation.

25 Referee Comment: 8. Section 4.1 - this seems like background material to me. You didn't really do anything new here except compile some data available in databases. Again, the NASS data is pretty weak for these trends, especially trends reported as if they are completely accurate. This is qualitative data at best.

Author Response: The referee is correct that in section 4.1 we present information compiled from data available in databases. 30 This section represents a tremendous effort that mainly yielded qualitative interpretations. However, we believe that this section is sufficiently important to include in the paper as it further highlights the poor documentation and availability of drainage data in the United States, a point we wish to emphasize as we believe it has important policy implications. These

data need to be better collected and archived so that researchers, policymakers and managers can easily access this information instead of gleaning information from scanned copies of mid-20th century census reports. Presenting the incomplete, yet best available information regarding histories of drainage and cropping practices in our four study basins bolsters our conclusion that the most parsimonious explanation for reductions in watershed storage are caused by drainage

5 practices for corn-soy agriculture (p. 23, lines 7-9). In any case, we have shortened this section to the extent possible, without eliminating any of the key points.

Referee Comment: 9.Page 6, lines 16-21 - it is a stretch to cite Figure 5 to discuss cyclicity. It is not obvious in the figure. suggest that the authors find a way to make it visible or drop from the text.

**Comment [SK34]:** No change made to manuscript

Author Response: Having shown and discussed this figure with many people over the past few years, none have questioned the cyclicity that emerges over the past ~35 years in the MRB with a 10-12 year wavelength. Granted, there are only 3 cycles, but the pattern seems sufficiently clear to at least mention. We agree with the reviewer that this observation does not substantially impact our results or interpretations substantially, but believe it is worth pointing out for future research.

Referee Comment: 10. I found the data is Figure 6 to be the most compelling in the paper. Are the deviations consistent with the imposed breakpoints?

**Comment [SK35]:** No change made to manuscript

Author Response: The deviations between P and Q coincided well with the breakpoints identified from the CWT and

20 reported in Table 3. Regardless, the results of all statistical tests are not sensitive to breakpoints spanning four decades (p. 9, lines 26-26; Supplement Table S1)

Referee Comment: 11. Section 4.2.3 - the use of daily scale in this multi-year analysis is inappropriate. The results are very weak and do not really add to your argument. What about the "flashiness index"? There is no real link of this section to your

25 main issues and I would suggest dropping this section.

**Comment [SK36]:** See changes made in this document, p. 41

Author Response: We briefly discuss daily streamflow metrics in this multi-year analysis to demonstrate that streamflows are changing across multiple scales. We believe this point is sufficiently important to include as we expect that the broader water sciences community (namely aquatic ecologists and geomorphologists) are interested in streamflow changes at sub-

30 annual and sub-seasonal scales. Flow flashiness is important for aquatic organisms, nutrients, sediment transport and river channel change. The changes we document, especially for the May-June time period, are large. We are further investigating

these daily-scale changes in other ongoing research. Nevertheless, in response to the reviewer's concern we will reduce the content of this section and better link these results to the main questions and key points in the revised manuscript.

Referee Comment: Overall, the paper adds to the body of research that already exists on the topic. The topic is not new and it has been evaluated by many authors previously but there are some methods and techniques used and the issue of worthy of attention. I am recommending the paper be accepted with major revisions.

**Section 1.d - Author Response to Anonymous Referee #4**

10      First we would like to thank Anonymous Referee #4 for reviewing our manuscript. We are happy to incorporate specific changes in the revised manuscript based on the referee's suggestions, and would benefit from further clarification of general comments made by the referee. Original comments by the referee are denoted by "Referee Comment" and our responses are denoted by "Author Response".

15  Referee Comment: This paper treats an interesting topic: the effect of human activities on the precipitation-runoff patters in the Midwestern United States. In the last century, the land transformation from natural to agriculture and urban areas seriously affected changes in the hydrology of the study area. The results suggested that storage has decreased in intensively drained and cultivated basins by 30%-200% since 1975, but increased by 30% in the less agricultural basin. This has amplified the streamflow response to precipitation increases in the Midwest. While the results are quite interesting some
20  important information is obscured or not well described:

Author Response: It is difficult for us to provide a thorough reply without more information from the reviewer regarding the relevance of each of these items to our analysis and interpretations. References to page/line numbers would have also been helpful.

Referee Comment: - infiltration and hydraulic soil properties (e.g. spatial data)

Author Response: We included descriptions of the soils in each of the watersheds in a previous version of this paper (reviewed in HESS), but found that it did not substantially contribute to the analysis and interpretations. Our study evaluates
30  precipitation-runoff patterns at very large scales. Soil types vary considerably within our watersheds, which largely precludes any meaningful quantitative analysis of if/how soil properties themselves influence the hydrologic patterns we

**Comment [SK37]:** See changes to text on page 23 in this document in response to AR#4's comment

observe.  However, in response to the reviewer's concern we will briefly describe soils in each of the watersheds in Section 2 in the revised manuscript and include the following statements:

"Soils in the Minnesota River Basin consists of organic rich, but poorly drained mollisols with a very small area consisting of alfisols and entisols (Stark et al., 1996). The Illinois River basin is generally dominated mollisols, containing around 1% organic matter and generally of low to very low permeability, with some presence of more permeable alfisols and entisols (Arnold et al., 1999; Groschen et al., 2000).  The dominant soil orders found in the Red River of the North basin include mollisols and alfisols with some areas underlain by entisols and histosols (Stoner et al., 1993). In the Chippewa River basin, alfisols and spodosols are most prevelant, with occasional pockets of entisols, mollisols, and histosols (Hartemink et al., 2012; Soil Survey Staff, NRCS)."

Regarding infiltration, no quantitative measurements exist. Models of infiltration are inadequate to produce reliable information at these very large scales. For that reason we have not discussed infiltration in our study.

Referee Comment: - water storage capacity of ditches (what kind of ditches are these? Only surface drainage system? What about the sub-surface drainage network?)

Author Response: In this paper, we provide all available information on surface and sub-surface drainage. In several locations throughout the manuscript we discuss the fact that very little quantitative information is available for the sub-surface network. We do, however, report county-level drainage data from five US Census of Agriculture reports. These reports include both surface and subsurface drainage, but exclude private lands less than 500 acres; these data likely underestimate actual drainage extent considerably, as the majority of farms in Minnesota, Illinois, North Dakota, and Wisconsin are less than 500 acres (p. 6, line 16; p. 11, lines 21-23). Qualitatively we don't have any reason to believe that the surface water storage capacity of ditches in these watersheds has changed substantially.

Referee Comment: - methodology to recognize/map ditches (the authors highlighted some issues in the underestimation of their extent; is this issue critical for the suitability of the final results?)

Author Response: We believe that inaccurate and incomplete public information on agricultural drainage systems is a critical issue that limits scientific and management understanding of how drainage influences streamflow regimes and nutrient transport across scales (especially in the Mississippi River Basin). We discuss this point several times in the paper. While we are cognizant of, and transparent about, these limitations, we have specifically chosen analyses that circumvent the need for such information to the extent possible (such as the water budget). So, as we discuss in the manuscript, we believe better

**Comment [SK38]:** No change to manuscript

**Comment [SK39]:** No change to manuscript

constraints could be obtained on the impact of artificial drainage at large spatial scales if/when better information on drainage practices is available.

Referee Comment: - surface runoff related to the different agriculture practices through years.

> **Comment [SK40]:** No change to manuscript

Author Response: Surface runoff is integrally linked to infiltration. As we stated above, no quantitative information is available to constrain this process.

Referee Comment: Also, some recent relevant papers related to the drainage ditch role in water storage capacity, effects of

10   the changes in drainage ditches density and agricultural practices on water storage capacity are missed in the literature review.

> **Comment [SK41]:** No change to manuscript

Author Response: There is an immense body of literature on agricultural tile drainage, and we have tried to review papers most relevant to our study. Further elaboration on tile drainage could detract from the main points of our paper (p. 3, lines

15   28-32). If there is a specific paper on the topic that the referee would like to see us reference, please let us know.

Referee Comment: Unfortunately, because of the above critical issues, the paper needs to be restructured adding extra info to provide a more clear view of the processes involved in the study area.

20   References

[revised manuscript text omitted]

**Comment [SK59]:** Text and figure changed according to minor point by referee Livneh. Delet figure above and including figure on below.

with the horizontal dashed lines, which represent a time when the percent of the total acres harvested for the three commodity groups exceeded 60% in RRB and CRB, where hay and small grains have historically dominated, and 75% in the MRB and IRB, where corn and soybeans have historically dominated. We acknowledge that these breakpoints do not consider the actual extent of soybeans, which is assumed to be a surrogate approximation for area of drained croplands.

5 Soybean coverage is much higher for both MRB and IRB compared to RRB and CRB even before 1955. Considering the large proportion of the MRB and IRB watersheds cultivated for soybeans in the early 1950's combined with extensive (20-25%) drainage by 1940 and 1950 (Fig. 2), this suggests streamflow changes generally occurred after both precipitation and LCT changes.

We observe minimal changes in the energy of the annual and inter-annual precipitation signal for any basins during 10 the period of record, and therefore could not identify the timing of precipitation change in any basin using CWT (Fig. 4). However, Figure 4 displays significant increases in the annual and inter-annual energy of the basin outlet streamflow signal around 1975, 1980, and 1995 for the IRB, MRB, and RRB respectively, while the CRB does not exhibit any striking changes in energy throughout the period of record. All decadal energy shifts in the precipitation signals are clearly translated into the decadal energy of the streamflow signals for all four basins (Fig. 4). The observed correlation between the decadal energy 15 changes in streamflow and precipitation signals together with the lack of any significant correlation between their energies at the annual scale may signal the importance of factors other than precipitation, here artificial drainage, to streamflows in the MRB, RRB, and IRB at the annual scale.

In all basins, the timing of precipitation change coincided with or preceded streamflow breakpoints based on PwLR (Table 3). Similar temporal coincidence of precipitation and streamflow breakpoints in contrast to the LCT and streamflow 20 breakpoints may suggest that streamflow changes are tightly coupled with precipitation changes. However, that interpretation fails to account for the potential effects of drainage, which could amplify the streamflow response to precipitation.

[Figure]

**Figure 4. Continuous Wavelet Transform (CWT) energies for monthly volumetric streamflow (Q) and precipitation (P) time series.**

Comment [SK60]: Figure changed according minor point by referee Livneh . Deleting figure on left and including figure on right.

[revised manuscript text omitted]

Comment [SK63]: Figure changed according minor point by referee Livneh. Deleting figure on left and including figure on right.

Comment [SK64]: Figure changed according minor point by referee Livneh. Deleting figure on left and including figure on right.

**Figure 8. Log-log empirical quantiles of joint PDF plots of monthly streamflow (Q) versus monthly precipitation (P) volumes for each river basin during the pre-period (blue: 1935-1974) and post-period (red: 1975-2013); bulls eye shading represent the 0.1 (dark), 0.6 (medium), and 0.9 (light) confidence intervals.**

**4.2.3 Daily scale changes of streamflow**

At the daily scale, we found an increase in the magnitude of streamflow change (hydrograph slopes) for both the daily rising limbs (dQ/dt>0) and falling limbs (dQ/dt<0) of the hydrographs for RRB, MRB, and IRB outlet gauges, suggesting an increase in "flashiness", or daily rate of change, of the hydrologic response (Fig. 9).  Figure 9 shows a slight decrease in the post-period curve for the CRB, indicating that the rising limb and falling limb flows may actually be less "flashy" in recent times than in the past. May-June is approximately the start of the growing season for soybean and corn and it is the time that tiles are most active, as this time of year usually corresponds to high monthly rainfall, high antecedent moisture conditions from spring snowmelt, and lower ET rates than the peak growing season due to lower crop water demands, and air temperatures that precede the annual peak.

[Figure]

**Figure 9. Daily streamflow change exceedance probabilities, where daily (dQ/dt>0) characterizes rising limb flows and daily (dQ/dt<0) characterizes falling limb flows.  Study basin acronyms are defined as follows: Red River of the North basin (RRB), Minnesota River basin (MRB), Chippewa River basin (CRB), and Illinois River basin (IRB).**

**Comment [SK65]:** Figure changed according minor point by referee Livneh. Deleting figure on left and including figure on right.

**Comment [SK66]:** Addresses Anonymous Referee #2 comment

**Comment [SK67]:** Text and figure changed according to minor point by referee Livneh. Delet figure at left and including figure at right.

**4.3 Hydrologic budgets suggest declining watershed storage in agricultural basins**

While time series and statistical analyses reveal useful insights regarding the timing, magnitude, and significance of precipitation and streamflow changes, as well as provide a qualitative indication of whether or not changes in precipitation and streamflow may be correlated and proportional, they cannot fully deconvolve or attribute the influence of artificial drainage and climate on streamflows (Harrigan et al., 2014). Therefore, we calculate water budgets for each basin as a tool to understand whether the observed changes in precipitation are large enough to account for the changes in streamflow, and if there is more or less watershed storage in recent times than in the past (Healy et al., 2007).

Table 4 reports the calculated average annual water budget terms – precipitation, streamflow, evapotranspiration, and change in storage – during the periods before and after the 1974/1975 and LCT breakpoint using raw and conservative (reduced by 17% in JJA) estimates of $ET_a$. We find that regardless of the breakpoint or raw vs. conservative estimates of $ET_a$ there is a net reduction in water stored in soil, groundwater, and/or lakes, wetlands, or reservoirs between the pre period and post period in the MRB, RRB, and IRB (Table 4). The most parsimonious explanation for this reduction in water storage is the systematic removal of wetlands and lowering the water table, accomplished through tile drainage installation and expansion.

The CRB, which is not intensively drained (Fig. 2) and has experienced little change in crop type (Fig. 3), has been subject to an increase in precipitation, but does not exhibit an increase in runoff (Table 4), consistent with Figs. 8 & 9b. The overall trends in the CRB water budget indicate that water storage may have actually increased slightly between the pre-period and post-period, which could be accomplished through increased soil moisture, groundwater recharge, or reservoir storage in recent times.

Using conservative estimates of summer $ET_a$ the change in storage term has decreased by about 200%, 100%, and 30%, in the MRB, IRB, and RRB from the pre-LCT-period to post-LCT-period. In the CRB, change in storage has increased by roughly 30% from 1935-1974 to 1975-2011. These results are consistent with our hypothesis that increases in artificial drainage in the MRB, RRB, and IRB necessarily change how precipitation is transformed into streamflow and that increases in precipitation alone cannot explain changes in streamflow in these basins. Without pervasive artificial drainage in the CRB, while precipitation has increased slightly, flows have not changed, likely due to increases in soil moisture, groundwater, and/or lake, wetland and reservoir storage. Seasonal changes in storage shown in Fig. 10 suggest that soil moisture, groundwater, and/or lake, wetland, and reservoir storage in the spring and summer is negative, suggesting not enough P given $ET_a$ to produce observed flows, and positive in the fall suggesting more P and $ET_a$ than necessary to produce observed flows and thus an increase in storage during the fall.

The Red River of the North and Minnesota River basins have some of the poorest drained soils of the Upper Midwest and historically grew more hay and small grains than the other basins (Fig. 3). The introduction of artificial drainage combined with the replacement of hay and small grains with soybeans and the lack of major dams and municipal

and industrial water use, has resulted in pronounced streamflow amplification in response to land use and climate changes in the RRB and MRB relative to the IRB and CRB (Fig. 4). Additionally these two basins have seen greater changes in annual and even monthly precipitation (Figs. 7 and 8). However, the extensively drained Minnesota River Basin has seen the largest increases in flow and largest decrease in watershed storage for relatively similar climatic change to the IRB and RRB, and this is likely because of the high degree of watershed hydrologic alteration and connectivity from drainage and lack of other anthropogenic water uses.

[Figure]

**Figure 10. Average monthly (January – December) ds/dtchange in basin soil moisture, groundwater, and/or reservior storage (ds/dt), calculated after LCTland cover transition (LCT) years (see Table 3 for Illinois River basinIRB, Minnesota River basinMRB, and Red River of the North basinRRB LCT years), and after 1975 for Chippewa River basinCRB 
[revised manuscript text omitted]

| **Minnesota River basin** MRB | 1978/1979 | 1958/1959 | 1967/1968 | No change | 1980 |
| **Illinois River basin** IRB | 1961/1962 | 1981/1982 | 1981/1982 | No change | 1975 |
| **Chippewa River basin** CRB | 2009/2010 | 1995/1996 | 1995/1996 | No change | No change |

**Table 4.** Observed average annual precipitation (P), flow (Q), evapotranspiration (ET) and storage $\left(\frac{dS}{dt}\right)$ depths (cm y$^{-1}$) for each basin during the pre-period (a) and post-period (b) split by 1974/1975 (1) and land cover transition (LCT) (2) breakpoints.

| | | Years | $P_{mean}$ (cm y$^{-1}$) | $Q_{mean}$ (cm y$^{-1}$) | $ET_{mean}$ (cm y$^{-1}$) | $\frac{dS}{dt}_{mean}$ (cm y$^{-1}$) |
|---|---|---|---|---|---|---|
| Minnesota River basin B | 1a | 1935-1974 | 65.1 | 7.2 | 60.9 | -3.0 |
| | 1b | 1975-2011 | 70.0 | 13.4 | 64.2 | -7.5 |
| | 2a | 1935-1978 | 64.8 | 7.0 | 60.6 | -2.8 |
| | 2b | 1979-2011 | 71.0 | 14.4 | 65.0 | -8.4 |
| | 1a† | 1935-1974 | | | 55.6 | 2.3 |

| Basin | Scenario | Period | | | | |
|---|---|---|---|---|---|---|
| | 1b† | 1975-2011 | | | 58.7 | -2.0 |
| | 2a† | 1935-1978 | | | 55.4 | 2.4 |
| | 2b† | 1979-2011 | | | 59.3 | -2.7 |
| Red River of the North basin | 1a | 1935-1974 | 53.4 | 3.7 | 45.1 | 4.7 |
| | 1b | 1975-2011 | 57.7 | 6.7 | 47.4 | 3.5 |
| | 2a | 1935-2003 | 54.5 | 4.6 | 45.6 | 4.4 |
| | 2b | 2004-2011 | 63.3 | 10.1 | 51.6 | 1.5 |
| | 1a† | 1935-1974 | | | 41.1 | 8.6 |
| | 1b† | 1975-2011 | | | 43.3 | 7.6 |
| | 2a† | 1935-2003 | | | 41.6 | 8.4 |
| | 2b† | 2004-2011 | | | 47.4 | 5.8 |
| Illinois River basin | 1a | 1939-1974 | 90.5 | 27.3 | 73.2 | -10.0 |
| | 1b | 1975-2011 | 95.2 | 33.0 | 75.1 | -13.0 |
| | 2a | 1939-1961 | 89.5 | 25.9 | 72.8 | -9.3 |
| | 2b | 1962-2011 | 94.4 | 32.2 | 74.8 | -12.5 |
| | 1a† | 1939-1974 | | | 66.9 | -3.7 |
| | 1b† | 1975-2011 | | | 68.7 | -6.6 |
| | 2a† | 1939-1961 | | | 66.5 | -3.0 |
| | 2b† | 1962-2011 | | | 68.4 | -6.1 |
| Chippewa River basin | 1a | 1935-1974 | 80.0 | 29.7 | 61.8 | -11.5 |
| | 1b | 1975-2011 | 82.1 | 29.8 | 62.7 | -10.5 |
| | 2a | 1935-2009 | 80.8 | 29.6 | 62.1 | -11.0 |
| | 2b | 2010-2011 | 88.4 | 33.3 | 68.5 | -13.4 |
| | 1a† | 1935-1974 | | | 56.5 | -6.2 |
| | 1b† | 1975-2011 | | | 57.4 | -5.2 |
| | 2a† | 1935-2009 | | | 56.8 | -5.7 |
| | 2b† | 2010-2011 | | | 62.3 | -7.3 |

† 17% reduction in ET during summer months (JJA)

*Supplement of*

**Human amplified changes in precipitation-runoff patterns in large river basins of the Midwestern United States**

Sara A. Kelly et al.

*Correspondence to*: Sara A. Kelly (sara.kelly@aggiemail.usu.edu)

[Figure]

**Figure S1: Field land use and tile arrangement before (1937) and after (1952) tile installation (1948) near Mapleton, MN (adapted from Burns, 1954); aerial photograph flown in spring 2013 shows the modern tile pattern remains relatively unchanged with a corn-soybean crop rotation (2009-2010), from the Cropland Data Layer (USDA NASS, 2013).**

Field Code Changed

[Figure]

**Figure S2. Seasonally averaged long term daily Parameter elevation Regression on Independent Slopes Model (PRISM) precipitation means (1981-2010) across the Upper Midwest: spring (MAM), summer (JJA), autumn (SON), and winter (DJF); USGS gage locations for each study basin (Table 1) indicated by open triangles (PRISM Climate Group, 2004). Study basin acronyms are defined as: Red River of the North basin (RRB), Minnesota River basin (MRB), Chippewa River basin (CRB), and Illinois River basin (IRB).**

Supplement of Section 3.2 - Climate records: precipitation and evapotranspiration

Comparison of monthly precipitation total reported as an average depth (cm) from Parameter elevation Regression on Independent Slopes Model (PRISM) , used in this study, and Livneh et al. (2013) (L13) for each watershed is shown in Figure S3. If PRISM and L13 precipitation depths were equivalent in every month, then all points would plot on the 1:1 line. On average (1935-2011) the difference between the two monthly precipitation datasets is 1% for each study watershed.

[Figure]

**Figure S3. Spatially averaged, total monthly (cm) precipitation (1935-2011) for each watershed from Parameter elevation Regression on Independent Slopes Model (PRISM Climate Group, 2004) and Livneh et al. 2013 (L13) plotted with 1:1 line.**

Figure S4 shows a comparison of monthly (March-November during 2001-2011) $ET_a$ estimates produced by Livneh et al. (2013) (L13) with $ET_p$ estimates (available from: http://agwx.soils.wisc.edu/uwex_agwx/sun_water/et_wimn) produced following the methods of Diak et al. (1998) (D98) for a location in the Minnesota River basin (MRB), 44 N, 94 W and the Chippewa River basin (CRB), 45.2 N, 91.6 W. On average, the estimates of $ET_a$ are 19% (raw) and 26% (17% reduction in JJA $ET_a$) lower than estimates of $ET_p$ in the MRB, and 16% (raw) and 24% (17% reduction in JJA $ET_a$) lower than estimates of $ET_p$ in the CRB.

[Figure]

**Figure S4. Monthly (March-November) average daily (mm d⁻¹) estimates of ET$_p$ following methods of Diak et al., 1998 (D98) versus estimates of ET$_a$ from Livneh et al., 2013 (L13) during 2001-2011.**

Figure S5 shows average monthly ET$_a$ from Livneh et al. (2013) compared against four AmeriFlux sites near the study watersheds (Table 2) as well as data from Bryan et al. (2015). In general, the L13 data show an earlier peak in ET$_a$ for the cropland sites in Rosemount, MN and Bondville, IL, and overestimate average annual ET$_a$ by 17% (raw) and 7% (17% reduction in JJA) for Bondville and 14% (raw) and 5% (17% reduction in JJA) for Rosemount. The L13 data overestimate ET$_a$ at Willow Creek, WI (broadleaf deciduous forest) by as much as 31% (raw) and 19% (17% reduction in JJA) annually, and underestimate ET$_a$ at Brookings, SD (grassland) by 29% (raw) and 34% (17% reduction in JJA) annually.

[Figure]

**Figure S5. Average monthly evapotranspiration rate (mm d⁻¹) at four AmeriFlux sites (see Table 2) compared to modeled evapotranspiration rates used in this study (L13 & L13-JJA) and Bryan et al. 2015.**

The L13 ET$_a$ estimates were calculated in VIC using the Hansen et al. (2000) static global vegetation classification, and did not consider artificial drainage, Therefore, the dominant mechanism for losing soil water in May and June is expected to be through ET$_a$ loss according to the L13 estimates. In contrast, ET$_a$ losses in May and June at Ameriflux sites are relatively low since crops are absent or very young and soil water likely drains primarily via artificial drainage. We expect that the effects

of drainage influence $ET_a$ during the peak growing season as well. Because drainage improves crop growing conditions early in the growing season, late growing season $ET_a$ may be higher in drained fields than undrained fields. This would be an interesting further line of study. Regardless, it seems reasonable that the L13 $ET_a$ estimates would seasonally mismatch the Rosemount and Bondville Ameriflux station $ET_a$ estimates, given the presence/absence of artificial drainage.

$ET_a$ estimates may dramatically underestimate Ameriflux $ET_a$ estimates in Brookings, SD due to differences in crop coefficients or misclassification of grasslands and croplands; corn has been found to have lower $ET_a$ rates than some grasses (Hickman et al., 2010). Due to the coarse resolution of the global vegetation input for the L13 VIC model, parts of southern Wisconsin appear to be misclassified as broadleaf deciduous forest instead of cropland. Some studies in the Great Lakes region report broadleaf deciduous forest to have slightly higher annual $ET_a$ rates than cropland (Mao and Cherkauer, 2009; Mishra et al., 2010). Likely of larger significance is that Livneh et al. (2013) and Maurer et al. (2002) do not suggest that they considered lake and wetland effects on evapotranspiration, which in the Great Lakes region can be significant (Bryan et al., 2015). Furthermore, the Hansen et al. (2000) global vegetation classification masks bodies of water, as the land cover input.

The fact that the L13 $ET_a$ estimates mismatch Ameriflux estimates seasonally provides assurance that the L13 $ET_a$ estimates are appropriate for testing our hypothesis. The lack of artificial drainage is what allows us to test whether factors beyond climate contribute to modern streamflow increases in the Midwestern US.

Comment [SK70]: Text addresses major point made by referee Livneh

[Figure]

**Figure S6. Annual, spatially averaged watershed precipitation and streamflow depths (cm) for each study basin.**

Table S1. Resulting p-values of 624572 statistical tests (t-test and Kolmogorov–Smirnov [KS]-test) comparing pre-period and post-period flow and precipitation based on the 1974/1975, piecewise linear regression (PwLR), and land cover transition (LCT) breakpoints for each basin (Table 3). P-values are highlighted based on their significance: bolded values are p-values with 95% confidence level or greater, grey values are p-values with less than a 95% confidence level, and black values are p-values where significance depends on the breakpoint. Italicized grey values reported for the CRB are not reliable because the post-period includes fewer than 10 years of data.

| Flow: t-test | | | Flow: KS-test | | | Precipitation: t-test | | | Precipitation: KS-test | | |
|---|---|---|---|---|---|---|---|---|---|---|---|
| **74/75** | **PwLR** | **LCT** | **74/75** | **PwLR** | **LCT** | **74/75** | **PwLR** | **LCT** | **74/75** | **PwLR** | **LCT** |

| | | | | | | | | | | | | | |
|---|---|---|---|---|---|---|---|---|---|---|---|---|---|
| **Chippewa River basinRB** | January | 0.341 | 0.893 | *0.846* | 0.653 | 0.958 | *0.902* | 0.278 | 0.097 | *0.214* | 0.223 | 0.050 | *0.082* |
| | February | 0.372 | 0.680 | *0.851* | 0.449 | 0.878 | *0.953* | **0.337** | **0.039** | *0.309* | 0.446 | 0.071 | *0.367* |
| | March | 0.566 | 0.871 | *0.525* | 0.219 | 0.749 | *0.205* | 0.188 | 0.369 | *0.574* | 0.234 | 0.348 | *0.700* |
| | April | 0.468 | 0.267 | *0.719* | 0.506 | 0.152 | *0.416* | 0.192 | 0.277 | *0.258* | 0.169 | 0.308 | *0.575* |
| | May | 0.826 | 0.485 | *0.264* | 0.482 | 0.311 | *0.622* | 0.933 | 0.374 | *0.187* | 0.906 | 0.697 | *0.445* |
| | June | 0.900 | 0.552 | *0.211* | 0.908 | 0.628 | *0.142* | 0.833 | 0.434 | *0.117* | 0.945 | 0.587 | *0.246* |
| | July | 0.706 | 0.775 | *0.308* | 0.584 | 0.893 | *0.606* | 0.463 | 0.609 | *0.358* | 0.567 | 0.794 | *0.360* |
| | August | 0.174 | 0.508 | *0.364* | 0.354 | 0.450 | *0.200* | 0.496 | 0.945 | *0.769* | 0.760 | 1.000 | *0.856* |
| | September | 0.517 | 0.990 | *0.723* | 0.357 | 0.958 | *0.654* | 0.286 | 0.912 | *0.752* | 0.657 | 0.925 | *0.654* |
| | October | 0.103 | 0.778 | *0.593* | 0.110 | 0.887 | *0.817* | **0.022** | **0.026** | *0.304* | 0.097 | 0.073 | *0.423* |
| | November | 0.240 | 0.894 | *0.713* | 0.337 | 0.887 | *0.902* | 0.510 | 0.905 | *0.806* | 0.375 | 0.944 | *0.874* |
| | December | 0.263 | 0.973 | *0.806* | 0.387 | 0.971 | *0.931* | 0.380 | 0.135 | *0.062* | 0.337 | 0.175 | *0.045* |
| | **Annual** | 0.499 | 0.793 | *0.340* | 0.721 | 0.918 | *0.291* | 0.243 | 0.571 | *0.295* | 0.246 | 0.764 | *0.614* |
| **Illinois River basinB** | January | 0.123 | 0.030 | 0.092 | 0.136 | 0.079 | 0.265 | 0.543 | 0.250 | 0.529 | 0.454 | 0.425 | 0.211 |
| | February | 0.355 | 0.082 | 0.184 | **0.353** | **0.043** | **0.216** | 0.224 | 0.108 | 0.764 | 0.433 | 0.359 | 0.899 |
| | March | 0.035 | 0.062 | 0.045 | 0.104 | 0.114 | 0.099 | 0.649 | 0.777 | 0.619 | 0.619 | 0.836 | 0.828 |
| | April | 0.174 | 0.438 | 0.158 | 0.335 | 0.479 | 0.353 | 0.780 | 0.832 | 0.588 | 0.883 | 0.947 | 0.638 |
| | May | 0.182 | 0.344 | 0.155 | 0.126 | 0.398 | 0.161 | 0.212 | 0.113 | 0.326 | 0.063 | 0.063 | 0.138 |
| | June | 0.105 | 0.077 | 0.280 | 0.082 | 0.071 | 0.117 | 0.798 | 0.742 | 0.845 | 0.811 | 0.643 | 0.954 |
| | July | 0.451 | 0.411 | 0.525 | 0.518 | 0.436 | 0.614 | 0.453 | 0.585 | 0.214 | 0.585 | 0.519 | 0.443 |
| | August | 0.090 | 0.249 | 0.212 | 0.181 | 0.508 | 0.259 | 0.054 | 0.408 | 0.257 | **0.037** | **0.475** | **0.108** |
| | September | 0.004 | 0.062 | 0.009 | 0.003 | 0.111 | 0.041 | 0.511 | 0.465 | 0.118 | 0.685 | 0.728 | 0.113 |
| | October | 0.074 | 0.147 | 0.065 | 0.082 | 0.139 | 0.142 | 0.142 | 0.072 | 0.363 | 0.113 | 0.143 | 0.378 |
| | November | **0.034** | **0.007** | **0.041** | 0.075 | 0.008 | 0.074 | 0.023 | 0.004 | 0.109 | 0.053 | 0.013 | 0.045 |
| | December | **0.021** | **0.011** | **0.010** | **0.040** | **0.019** | **0.022** | 0.122 | 0.081 | 0.039 | 0.136 | 0.203 | 0.050 |
| | **Annual** | **0.011** | **0.019** | **0.017** | **0.012** | **0.018** | **0.020** | 0.086 | 0.075 | 0.085 | 0.117 | 0.141 | 0.183 |
| **Minnesota River Bbasin** | January | **0.000** | **0.000** | **0.000** | **0.001** | **0.000** | **0.000** | 0.096 | 0.520 | 0.182 | 0.369 | 0.672 | 0.338 |
| | February | **0.000** | **0.000** | **0.000** | **0.002** | **0.000** | **0.000** | 0.722 | 0.842 | 0.659 | 0.540 | 0.938 | 0.604 |
| | March | **0.007** | **0.005** | **0.002** | 0.089 | 0.055 | 0.074 | **0.017** | **0.394** | **0.060** | 0.107 | 0.515 | 0.148 |
| | April | **0.041** | **0.039** | **0.016** | **0.011** | **0.034** | **0.003** | 0.159 | 0.164 | 0.239 | 0.489 | 0.344 | 0.684 |
| | May | **0.000** | **0.001** | **0.000** | **0.001** | **0.001** | **0.001** | 0.716 | 0.366 | 0.469 | 0.807 | 0.657 | 0.636 |
| | June | **0.000** | **0.000** | **0.000** | **0.000** | **0.000** | **0.000** | 0.500 | 0.957 | 0.418 | 0.807 | 0.879 | 0.677 |
| | July | **0.002** | **0.007** | **0.000** | **0.021** | **0.020** | **0.003** | 0.300 | 0.107 | 0.151 | 0.351 | 0.230 | 0.223 |
| | August | **0.008** | **0.012** | **0.000** | **0.017** | **0.040** | **0.001** | 0.239 | 0.641 | 0.133 | 0.097 | 0.806 | 0.064 |
| | September | 0.017 | 0.062 | 0.001 | 0.106 | 0.225 | 0.015 | 0.224 | 0.077 | 0.242 | 0.334 | 0.112 | 0.261 |
| | October | **0.002** | **0.001** | **0.000** | **0.012** | **0.007** | **0.002** | 0.082 | 0.015 | 0.029 | 0.115 | 0.088 | 0.116 |
| | November | **0.001** | **0.000** | **0.000** | **0.006** | **0.000** | **0.001** | 0.262 | 0.418 | 0.380 | 0.260 | 0.519 | 0.445 |

| | | | | | | | | | | | | | |
|---|---|---|---|---|---|---|---|---|---|---|---|---|---|
| | December | 0.000 | 0.000 | 0.000 | 0.002 | 0.000 | 0.000 | 0.435 | 0.138 | 0.385 | 0.608 | 0.457 | 0.413 |
| | **Annual** | **0.000** | **0.000** | **0.000** | **0.001** | **0.001** | **0.000** | 0.033 | 0.072 | 0.011 | 0.098 | 0.199 | 0.051 |
| **Red River of the North basin**  | January | 0.000 | 0.000 | 0.005 | 0.005 | 0.000 | 0.013 | 0.117 | 0.169 | 0.412 | 0.112 | 0.105 | 0.368 |
| | February | 0.000 | 0.000 | 0.016 | 0.003 | 0.000 | 0.034 | 0.155 | 0.321 | 0.050 | 0.246 | 0.326 | 0.183 |
| | March | 0.006 | 0.012 | 0.171 | 0.011 | 0.005 | 0.217 | 0.050 | 0.108 | 0.021 | 0.062 | 0.247 | 0.054 |
| | April | 0.069 | 0.079 | 0.036 | 0.105 | 0.102 | 0.115 | 0.981 | 0.902 | 0.619 | 0.974 | 0.823 | 0.574 |
| | May | **0.016** | **0.003** | **0.004** | 0.059 | 0.013 | 0.014 | 0.321 | 0.039 | 0.046 | 0.312 | 0.129 | 0.186 |
| | June | **0.015** | **0.000** | **0.001** | **0.038** | **0.002** | **0.001** | 0.170 | 0.138 | 0.351 | 0.105 | 0.032 | 0.569 |
| | July | 0.000 | 0.000 | 0.001 | 0.001 | 0.000 | 0.005 | 0.288 | 0.251 | 0.886 | 0.244 | 0.418 | 0.943 |
| | August | 0.000 | 0.000 | 0.013 | 0.000 | 0.000 | 0.043 | 0.687 | 0.551 | 0.681 | 0.598 | 0.650 | 0.786 |
| | September | 0.002 | 0.000 | 0.024 | 0.002 | 0.000 | 0.084 | 0.094 | 0.036 | 0.047 | 0.009 | 0.013 | 0.081 |
| | October | **0.003** | **0.000** | **0.010** | 0.003 | 0.002 | 0.053 | **0.010** | **0.015** | **0.002** | **0.011** | **0.003** | **0.004** |
| | November | 0.000 | 0.000 | 0.001 | 0.000 | 0.000 | 0.003 | 0.409 | 0.560 | 0.918 | 0.270 | 0.341 | 0.943 |
| | December | 0.000 | 0.000 | 0.000 | 0.001 | 0.000 | 0.013 | 0.487 | 0.058 | 0.000 | 0.639 | 0.071 | 0.002 |
| | **Annual** | **0.000** | **0.000** | **0.000** | **0.000** | **0.000** | **0.005** | **0.019** | **0.004** | **0.009** | **0.010** | **0.008** | **0.027** |

15
- Addressed minor referee comments and made edits to figures, captions, abbreviations, word use, etc.

---

## Author Response (AR2)

**Section 1. Point-by-Point Replies to Referee Comments**

**Author Response to Anonymous Referee #2**

First we would like to thank Anonymous Referee #2 for re-reviewing our manuscript. Original comments by the
referee are denoted by "Referee Comment" and our responses are denoted by "Author Response".

**Referee Comment:** The present study associates observed changes in streamflow patters in four large basins of the
Midwestern Unites States to changes in precipitation, evapotranspiration and human interventions, particularly the increasing
cultivation of soybean and corn enhanced by artificial drainage. It analyses 79 years (1935 - 2013) of precipitation,
streamflow, artificial drainage, and cultivation data, to provide a extensive statistical time series analysis, identifying
breakpoint years that evidence the hypothesized changes. Additionally, the changes in basin storage are associated to
changes in climate or in land-cover using a simple water budget. The paper concludes that artificial drainage as part of large
agricultural development in the Midwestern US amplifies the changes in rainfall-runoff response because of an increased
hydrologic connectivity and a reduced storage in the basin.

This version is a minor revision of the original submission that I reviewed a few months ago; therefore, I do not see
significant changes to address successfully all of the referee comments. Most of the improvements have been made on the
form of the text and in a lesser extent to address the several criticisms on the document provided by the different reviewers. I
would be more keen to see a thorough revision of the manuscript than a justification of why such changes have not been
assimilated.

Particularly, with respect to those provided in my revision, I keep the main major comments as unsolved:

- It is not a stylistic request to divide results from discussion. It is indeed difficult to separate them because of the tendency to
confuse result description from interpretation. This is a skill necessary to develop in order to be objective at drawing
conclusions and remove speculations or interpretations that are not completely based on the current data and results.

**Author Response:** We agree with the referee that it is typically preferable to separate results and discussion, and this is the
style we generally follow in our publications. However, in this particular paper we found it difficult to follow that style as it
would increase redundancy and obscure clarity.  We hope the reviewer trusts our decision on this matter and accepts that
there are more than one styles of presenting results depending on the context (e.g., an interesting exposition of this topic is in
the book by Stephen B. Heard "The Scientist's Guide to Writing".) We note that HESS does not require authors to separate

results and discussion (as some other journals do), presumably in recognition of the fact that this format may not be appropriate for all papers.

In our paper we present three separate, but related research questions: (1) how have land use, climate, and streamflows changed during the 20th and 21st centuries; (2) what are the timing, time scales and times of year that changes are most prominent; and (3) can changes in climate alone explain changes in streamflow?. Questions 1 and 2 are closely related, but addressed using separate methods. Therefore, we present and discuss the results of these two questions together, but in separate sub-sections for land use and land cover (section 4.1), and climate and streamflow (section 4.2). In the revised manuscript we have created a separate introduction section, section 1.3, to clearly indicate to the reader where to find the main research questions, and reiterate these questions at the beginning of section 5. Because we present a wide variety of types of analyses to address these questions, it was very cumbersome – we tried – to take the reader through each set of results first in one section, and then to build up context again in a separate section to discuss the interpretation of each set of results. Within each sub-section we present a set of results, followed by discussion and interpretation. In an effort to modify the paper as recommended by the reviewer we also considered adding further sub-section headings for each set of results and discussion, but we feel that it breaks the paper up too much. Therefore, we feel that the clearest communication style for this paper is to present the results and a brief discussion of each result together in sub-sections of the paper. Additionally, in an effort to clarify the organization of the methods, results and discussion we have added brief statements at the beginning of sections 3 and 4 indicating where readers can find the methods and results that address each research question. This format allows us to make interpretations, discuss potential limitations, and conclude with implications from the combined results of our study in section 5.

We also agree with the reviewer that it is essential to minimize, and appropriately qualify, speculative statements. We have reviewed sections 4 and 5 and believe firmly that we have included appropriate caveats and qualifiers for all of our interpretations. If the reviewer has any specific concerns we would be happy to address them, but without more specific information regarding which statements elicit concern for the reviewer we are not able to respond directly regarding any particular instance.

We hope that the reviewer and editors understand our motivation for not adopting their suggested major reorganization of the paper and accept our writing style.

**Referee Comment:** - The study uses 7 metrics to analyze the streamflow regime that might be redundant as these are related to flow magnitude mainly. In the previous revision, I have recommended to include other hydrological indices that relate to

the different flow attributes (magnitude, frequency, duration, timing, and flashiness). The authors could perform a correlation analysis between the metrics, gauges, and catchments, or a Principal Component Analysis to either demonstrate that the indices they are using are indeed independent, or to identify more informative hydrological indices that characterize streamflow more comprehensively.

I judge the paper could be suitable for publication after these revisions.

**Author Response:** We chose to focus our analysis on metrics of magnitude, frequency and flashiness over multiple spatial and temporal scales. While we agree with the reviewer that it could be useful and insightful to analyse additional streamflow

10 metrics, we believe we have captured the most relevant metrics to answer our questions of interest. The 7 streamflow metrics we have presented in Figure 5 cover a wide range of flow attributes that characterize parts of the flow regime. Most of these attributes relate to magnitude, but occurrence of high and extreme flow days relate to frequency. These seven metrics are commonly used in hydrology and in fact comprised the dominant analysis in the widely cited paper of Novotny and Stefan, 2007. However, we go far beyond presenting these metrics and, in fact, already present many of the types of analyses the

15 reviewer requests. Namely, the wavelet analysis (Fig. 4) evaluates changes in frequency. Figure 6 evaluates changes in cumulative rainfall-runoff patterns. Figures 7 and 8 evaluate changes in monthly rainfall-runoff patterns. Furthermore, the exceedance plots of rising and falling limb flows (Fig. 9) demonstrate changes in flow flashiness. We chose to present these analyses because we believe they are the most meaningful for addressing our research questions and set the context for our water budget. We have not explicitly captured changes related to flood timing or duration in this paper and believe this type

20 of analysis would be better suited as the focus of a separate paper.

**Section 2. Summary of Major Manuscript Changes based on Referee Comments**

- Added sub-section heading 1.3 to introduction in effort to clearly indicate the research questions and improve organization
25
- Replaced land use land cover with LULC in several instances
- Added text at the beginning of section 4 to explain the organization of the results and discussion section
- Re-numbered/re-named sub-section headings in section 4 in effort to improve organization
- Added text at the beginning of section 5 to reiterate our research questions before summarizing the main findings of our study
30
- Note: Abstract and Supplement remain unchanged

[revised manuscript text omitted]

---

## Author Response (AR3)

**Section 1. Point-by-Point Replies to Referee Comments**

- Not applicable this round

**Section 2. Summary of Minor Manuscript Changes before Final File Upload**

5

- Changed spelling of gaging to gauging in Table 1 to be consisting with spelling throughout
- Made minor changes to figures to improve readability and comply with HESS figure guidelines
- Added a data availability statement (see Section 6 in latest version of manuscript)

10

**Section 3. Revised Manuscript with Edits as Track Change**

- Not applicable this round